# Controlling Forgetting with Test-Time Data in Continual Learning

## Abstract

Foundational vision-language models have shown impressive performance on various downstream tasks. Yet, there is still a pressing need to update these models later as new tasks or domains become available. Ongoing Continual Learning (CL) research provides techniques to overcome catastrophic forgetting of previous information when new knowledge is acquired. To date, CL techniques focus only on supervised training sessions. This results in significant forgetting yielding inferior performance to even the prior model zero shot performance. In this work, we argue that test-time data hold great information that can be leveraged in a self-supervised manner to refresh the model's memory of previously learned tasks and, hence, greatly reduce forgetting at no extra labeling cost. We study how unsupervised data can be employed online to improve models' performance on prior tasks upon encountering representative samples. We propose a simple yet effective student-teacher model with gradient-based sparse parameter updates and show significant performance improvements and reduction in forgetting. This could alleviate the role of an offline episodic memory/experience replay buffer.

## 1 Introduction

Foundation models in computer vision have shown impressive performance on various downstream tasks and domains, which renders them a key building block of various solutions, including generative vision language models Li et al. (2022); Chen et al. (2023); Bommasani et al. (2021). However, naively adapting pretrained models to changes in data distribution or new tasks faces the well-known catastrophic forgetting phenomena McCloskey & Cohen (1989), where new learning sessions interfere with what a model has previously acquired. To overcome catastrophic forgetting, Continual Learning (CL) has emerged as a branch of machine learning to enable models to continuously adapt to evolving distributions of training samples or supervision signals over time. A variety of approaches have been proposed to mitigate catastrophic forgetting, such as regularization-based methods Kirkpatrick et al. (2017); Maltoni & Lomonaco (2019); Schwarz et al. (2018), external memory approaches Lopez-Paz & Ranzato (2017); Li & Hoiem (2017), and dynamic model architecture techniques Shin et al. (2017); Singh et al. (2024). Most works focus on training models from scratch, which might fail with large pretrained models. The rise of foundation models has fueled interest in combining CL with the strengths of pretrained models. Han et al. (2021); Radford et al. (2021); Ridnik et al. (2021); Caron et al. (2021); Oquab et al. (2023); Radford et al. (2021).

Despite the increased attempts to efficiently improve foundational models performance on new streams of data Ermis et al. (2022); Pelosin (2022); Wang et al. (2022e); Smith et al. (2023); Janson et al. (2022); Zhou et al. (2023); Zhang et al. (2023a); Wang et al. (2022b); Ding et al. (2022); Goyal et al. (2023); Wang et al. (2022d), forgetting is still a significant problem in applications of continual learning Wang et al. (2024); Prabhu et al. (2023). Continual learning systems often process large volumes of unsupervised data throughout their lifecycle. We argue they must learn continuously, regardless of supervision. However, most works focus on supervised training, leaving models static during testing.

Consider an embodied agent with a Vision Language Model (VLM) that must handle new objects, layouts, or skills while still recognizing previously learned tasks during evaluation. To overcome catastrophic forgetting and accumulate knowledge, we propose leveraging test-time data to reinforce the model's understanding of prior tasks. Test-time data reflects the distribution most relevant to the

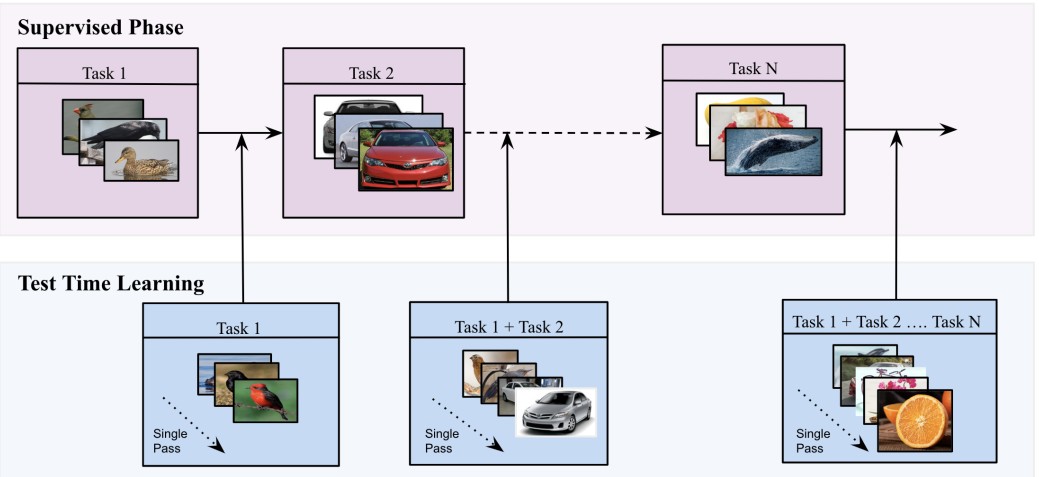

Figure 1: An illustration of our proposed Continual Learning with Interleaved Test Time Learning. Following each session of supervised learning, the model is deployed to adapt in an unsupervised setting. It can encounter data distributions encompassing all previously encountered tasks or sessions. The model adapts to the classes of the current task while trying to minimize the forgetting on all the classes of previously seen tasks.

agent's tasks, while past data never encountered during deployment can be forgotten to prioritize performance on frequently encountered scenarios.

We consider a scenario where a model undergoes continual supervised training, with unsupervised data available during deployment between training phases, offering a chance to mitigate forgetting. To ensure practical computational overhead, we constrain unsupervised adaptation to an online approach. Additionally, stringent data privacy constraints during deployment Verwimp et al. (2023) require online algorithms that discard samples immediately after processing.

Test-Time Adaptation (TTA) Sun et al. (2020) and Continual Test-Time Adaptation (CoTTA) Wang et al. (2022a) are related research areas that focus on leveraging test-time data for dynamic model adaptation. These areas focus on adapting the model towards unknown distribution shifts using test-time data, while our formulation aims to use test-time data to control the model forgetting without any assumption of distribution shifts from training to test data.

To the best of our knowledge, we are the first to explore how test-time data can be leveraged in a continual learning setting to reduce forgetting. We consider the foundation model CLIP Radford et al. (2021) for our experiments since it has been shown to encompass an extensive knowledge base and offer remarkable transferability Rasheed et al. (2023); Pei et al. (2023). It undergoes through supervised and unsupervised sessions, leveraging the unsupervised data to control forgetting.

We propose an effective approach based on student-teacher models with sparse parameter selection based on gradient values. Student and teacher models suggest labels for test data, and the predictions from the most confident model are used to update the student model, where the teacher is updated in an exponential moving average, adding a stability component to the learning process. We show that such a simple approach achieves significant improvements on all studied sequences. Our approach is stable in class incremental learning (CIL), especially in the challenging setting where no replay buffers are used, which in many cases can be a critical bottleneck.

Our contributions are as follows: 1) We propose a new setting for continual learning where test-time data can be leveraged, especially in the challenging scenario of CIL-CL. 2) We investigate different baselines for this setting. 3) We propose a novel approach that illustrates the utility of test-time data in supervised continual learning and the significant reduction in forgetting without any external replay buffer.

In the following, we discuss the closely related work, Section 2 and present our setting, and our approach, Section 3 we evaluate our approach on various CL sequences, Section 4, perform ablations on different components of our approach, Section 5, put forth some limitations of our work, Section 6 and conclude in Section 7.

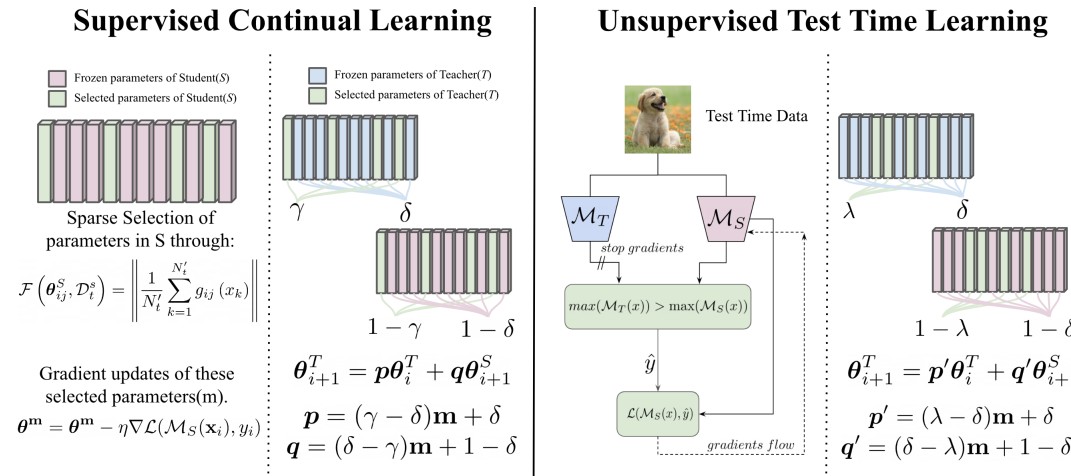

Figure 2: An illustration of our method DoSAPP. It utilises teacher-student ($\mathcal{M}_T$, $\mathcal{M}_S$) models respectively. During the Supervised Continual Learning phase, $\mathcal{M}_S$ undergoes sparse parameter selection through a gradient-based scoring function $\mathcal{F}$, followed by supervised training of these selected parameters $\theta^{\mathbf{m}} \in \theta^S$. After each gradient update step, $\mathcal{M}_T$ parameters, $\theta^T$, are updated through a weighted exponential smoothing based on the affine projection of the boolean mask: $\mathbf{m}$. The affine projections are controlled through dual momentum terms $\delta, \gamma$ for $\mathcal{M}_T$ and $\mathcal{M}_S$, respectively. Now both $\mathcal{M}_T$, $\mathcal{M}_S$ are deployed for the unsupervised test time learning where the $\mathcal{M}_S$ is adapted based on the "pseudo label groundtruth" generated from $\mathcal{M}_T$ - $\mathcal{M}_S$ logits comparison. Finally $\mathcal{M}_T$ model again undergoes weighted smoothing, with dual momentum terms $\delta, \lambda$ for $\mathcal{M}_T$ and $\mathcal{M}_S$ model respectively such that $\gamma < \lambda < \delta$. This 2 phase approach preserves the generalizations over previous knowledge along with adaptability on the latest task.

## 2 RELATED WORK

**Continual Learning:** It considers learning in an incremental manner where training data is received at various time steps (sessions). The typical problem is catastrophic forgetting McCloskey & Cohen (1989) of previously learned information. We refer to De Lange et al. (2021) for a survey on class incremental learning where different classes are learned at distinct sessions, a setting we consider in this work. Weight regularization methods Aljundi et al. (2018); Kirkpatrick et al. (2017) and functional regularization Li & Hoiem (2017); Asadi et al. (2023) direct the training to stay optimal for tasks of previous sessions via various regularization terms. Experience Replay French (1999) is usually deployed where samples of previous training session data are replayed during new sessions to reduce forgetting. In this work, we consider continual learning with limited or no replay. Our work is orthogonal to other continual learning methods and can be combined with any CL method in the supervised training sessions.

**Continual Learning from Pre-trained Models:** Due to the abundance of powerful pre-trained models Radford et al. (2021); Oquab et al. (2023); Brown et al. (2020), continual learning that begins with a pre-trained model is becoming a popular paradigm. Recent methods like Koh et al. (2022); Boschini et al. (2022) have utilized a Teacher-Student framework for knowledge distillation on previously seen tasks. However, these methods utilize an additional buffer to mitigate catastrophic forgetting. This often entails significant memory Zhou et al. (2022); Prabhu et al. (2023). Additionally, such methods often face an outdated logit problem, as the memory-stored logits are not updated to preserve information on previous tasks. Boschini et al. (2022) addresses this issue by updating logits stored in the past using task boundary information (e.g., input's task identity) during training, but it may not always be available, especially in task-free CL setups. However, foundation models Radford et al. (2021); Oquab et al. (2023) often have a reasonable initial performance on novel tasks, indicating some pre-existing knowledge relevant to these tasks. Zhang et al. (2023b) utilizes this property and preservers generic knowledge by modifying only a small set of parameters based on gradient scoring mechanism. However, this method also suffers from recency bias since the gradient scores are computed for the current task, and only those sparse parameters are updated based on current task scores. Moreover, none of the methods utilize test data in continual learning scenarios and leave a strong potential for self-supervised techniques to capture robust feature representations.

**Test Time Adaptation (TTA):** TTA has been extensively studied in recent years, focusing on adapting a pre-trained model based on test data. Typically, the goal is to enhance performance on the test data used for adaptation. However, our focus is on using this data to mitigate forgetting of past tasks. Various methods have been proposed for TTA, including those leveraging self-supervised learning Sun et al. (2020), batch normalization Nado et al. (2020); Vianna et al. (2024), entropy minimization Wang et al. (2020); Niu et al. (2023), and pseudo labeling Chen et al. (2022); Li & Hoiem (2017). It is important to note that our method is not merely a modification, a novel variation, or a combination of existing TTA approaches. Unlike typical TTA methods, which primarily address data corruption and show limited benefits when changes are restricted to label distributions, our approach leverages unsupervised data from previous tasks without requiring it to be corrupted to deliver its advantages.

**Continual Test-Time Adaptation:** Recent work has studied the setting of performing Online Test-Time Adaptation where the distribution of test-time data is changing over time Wang et al. (2022a). This is distinct from the proposed setting as we focus on the setting where the model is updated with supervised data while the test-time data is leveraged to control forgetting on the supervised tasks.

## 3 METHODOLOGY

We introduce a novel setting for continual learning that leverages test-time data, particularly in the challenging context of *Class Incremental Continual Learning* (CIL-CL). As depicted in Figure 1, this setting allows the deployed model to recover lost knowledge from distributions spanning all previously encountered tasks after each supervised learning session. The model adapts to the current task's classes while minimizing forgetting of earlier tasks' classes. Our proposed approach demonstrates how test-time data can significantly reduce forgetting in supervised continual learning, achieving this without relying on an external replay buffer.

### 3.1 SETTING

We consider a setting where a sequence of supervised datasets $[\mathcal{D}_1^s, \mathcal{D}_2^s, .....\mathcal{D}_T^s]$ drawn from different distributions are observed at incremental training sessions $t$ ranging from 0 to $T$, where $\mathcal{D}_t^s = (\mathbf{x}_i^t, y_i^t)_{i=1}^{N_t}$ is the $t$ incremental session with $N_t$ instances. Here the training instance $\mathbf{x}_i^t \in \mathbb{R}^D$ belongs to class $y_i \in Y_t$, where $Y_t$ is the label space of task/dataset at $t$ step. $Y_t \cap Y_{t'} = \phi$ for $t \neq t'$, where $t'$ is any other training session. During a given training session, $t$ data samples only from $\mathcal{D}_t^s$ can be accessed. CIL aims to progressively build a unified model encompassing all previously encountered classes. This involves gaining insights from new classes while retaining knowledge from previous ones. The model's performance is evaluated over all the seen classes $\mathcal{Y}_t = Y_1 \cup \cdots Y_t$ after each incremental task/dataset. Formally, the target is to fit a model $\mathcal{M}(\mathbf{x}; \boldsymbol{\theta}) : X \to \mathcal{Y}_t$ that achieves a minimal loss $\mathcal{L}$ across all testing datasets $\mathcal{D}_t^e$ :

$$\sum_{(\mathbf{x}_j, y_j) \in \mathcal{D}_1^e \cup \cdots \mathcal{D}_T^e} \mathcal{L}\left(\mathcal{M}\left(\mathbf{x}_j; \boldsymbol{\theta}\right), y_j\right) \tag{1}$$

where $\mathcal{L}(.,.)$ measures the difference between prediction and groundtruth label. $\mathcal{D}_t^e$ denotes a testing set of task $t$. Finally, $\theta$ denotes the model parameters.

After training is complete on each $\mathcal{D}_t^s$, the model is put into production until $\mathcal{D}_{t+1}^s$ becomes available for supervised training. Between supervised phases, an unsupervised dataset, $\mathcal{D}_t^u$, is observed corresponding to test-time data encountered in production. Note that this unsupervised data can be drawn from a different distribution than the supervised data, including the distributions of old supervised datasets/tasks. Our goal is to leverage this data to control the forgetting of the model by allowing online unsupervised adaptation. Figure 1 depicts our setting. Note that we evaluate our models on test datasets $\{\mathcal{D}^e\}$ that are distinct in terms of instances from those used during the self-supervised online adaptation phase to adequately measure model generalization.

We further note that although supervised phases may permit multiple passes through the data until convergence, it would be impractical to collect unsupervised data in production and then perform adaptation on it. We thus restrict the unsupervised phase to be in the online setting Sun et al. (2020); Jang et al. (2022); Cai et al. (2021). This is especially important in cases where data privacy is important e.g., assistant robot in a private smart home environment.

## 3.2 DoSAPP: Double Smoothing via Affine Projected Parameters

---

**Algorithm 1** DoSAPP algorithm for continual and test time learning

---

**Require:** $\mathcal{M}_S(\theta^S)$, CLIP loss: $\mathcal{L}(.,.,.)$, sparsity threshold $c$
1: $\theta^T = \theta^S$                                                     ▷ Initialize $\mathcal{M}_T(\theta^T)$ with $\mathcal{M}_S(\theta^S)$
2: **for** $t$ in tasks **do**
3:      $\theta^{\mathbf{m}} \leftarrow$ top-K(K=c) params from MLP layers of $\theta^S$ based on $\mathcal{F}$     ▷ Sparse Selection, Eq. 2
4:      **for** $(x_i, y_i)$ in $\mathcal{D}_t^s$ **do**
5:          $\theta^{\mathbf{m}} = \theta^{\mathbf{m}} - \eta \nabla \mathcal{L}(\mathcal{M}_S(x_i), y_i)$                       ▷ Take one SGD step
6:          $\theta_{i+1}^T = p\theta_i^T + q\theta_{i+1}^S$           ▷ Dual momentum for teacher EMA update, Eq 4
7:      **end for**
8:      Compute union of masks for all tasks seen so far $\mathbf{m}_u$       ▷ Start of Unsupervised Phase
9:      Select $\mathbf{m}_u$ params in $\mathcal{M}_S$
10:     **for** $x_i$ in $\mathcal{D}_t^u$ **do**
11:        $l_T = \max(\mathcal{M}_T(x_i), \dim = 1)$
12:        $l_S = \max(\mathcal{M}_S(x_i), \dim = 1)$
13:        **if** $l_T > l_S$ **then**
14:           $\hat{y} = \arg\max(\mathcal{M}_T(x_i))$
15:        **else**
16:           $\hat{y} = \arg\max(\mathcal{M}_S(x_i))$
17:        **end if**
18:        $\theta^{\mathbf{m}_u} = \theta^{\mathbf{m}_u} - \eta \nabla \mathcal{L}(\mathcal{M}_S(x_i), \hat{y})$               ▷ Take one SGD step
19:        $\theta_{i+1}^T = p'\theta_i^T + q'\theta_{i+1}^S$         ▷ Dual momentum for teacher EMA update, Eq 7
20:     **end for**
21: **end for**

---

We propose a simple yet effective method for continual test-time learning, Double Smoothing via Affine Projected Parameters, aka DoSAPP. Our approach combines two key components: **1) sparse and local updates:** to reduce forgetting, maintain generalization, and ensure efficient updates, and **2) teacher-student framework** to promote stability in online updates and minimize forgetting. In the continual test time learning, we can identify two distinct phases of learning, as outlined below.

### PHASE 1: CONTINUAL LEARNING SUPERVISED TRAINING WITH SPARSE SELECTED PARAMETERS

Our primary objective is to swiftly accumulate new knowledge without catastrophically forgetting the generic knowledge both at training and test time. To achieve this, we opt for updating only a small subset of selected parameters. It has been suggested by Zhang et al. (2023b) that for a generic pretrained model like CLIP and a given task, relevant parameters can be identified before training, and updating only those parameters would result in a reduced forgetting of previous knowledge. Further Geva et al. (2020) suggested that MLP blocks in a transformer model emulate key-value neural memories, where the first layer of MLP acts as memory keys operating as pattern detectors. This suggests that for updating knowledge of previously known "*patterns*", it might be sufficient to update only the first MLP layer parameters. Thus we limit candidate parameters to the first MLP layer parameters of each transformer block in the CLIP model Zhang et al. (2023b). From these candidate parameters of the first MLP layer of each transformer, we select top-K (K=c) parameters. This results in efficient training without loss of previously acquired knowledge as all other layers remain frozen.

Following Zhang et al. (2023b) we use the gradient magnitude of the loss w.r.t. the incoming data as a score of how relevant a parameter is, the larger the gradient magnitude the larger the expected decrease in loss after small changes to that parameter. We refer to the model being optimized as $\mathcal{M}_S$. Upon receiving supervised data, we first estimate the most relevant parameters, $\boldsymbol{\theta}^{\mathbf{m}}$ such that $(\boldsymbol{\theta}^{\mathbf{m}} \in \boldsymbol{\theta}^S)$.

$$\mathcal{F}\left(\boldsymbol{\theta}_{ij}^S, \mathcal{D}_t^s\right) = \left\| \frac{1}{N_t'} \sum_{k=1}^{N_t'} g_{ij}\left(x_k\right) \right\| \tag{2}$$

where $g_{ij}(x_k)$ is the gradient of the loss function ($\mathcal{L}(\mathcal{M}_S, x_k, y_k)$) regarding the parameter $\boldsymbol{\theta}_{ij}^S$ evaluated at the data point and its label $x_k, y_k \in \mathcal{D}_t^s$. The loss function $\mathcal{L}(\mathcal{M}_S, x_k, y_k)$ is the same CLIP loss, and the entire data is iterated once to compute the gradient score as given in Eq 2. Specifying the sparsity threshold ($c$), top-K (K=$c$) most relevant parameters are selected. We set $c = 0.1$ as shown in Zhang et al. (2023b). This results in a binary mask $\mathbf{m}$ where only selected parameters are updated, and others are masked out and kept frozen.

### TEACHER STUDENT FRAMEWORK

To ensure stability later during online updates and reduce forgetting, we utilize a Student-Teacher framework Tarvainen & Valpola (2017); Koh et al. (2022); Boschini et al. (2022) where the student model is denoted by $\mathcal{M}_S(\boldsymbol{\theta}^S)$ and the teacher model is denoted by $\mathcal{M}_T(\boldsymbol{\theta}^T)$.

During both train and test time, teacher model $\mathcal{M}_T$ parameters $\boldsymbol{\theta}^T$ move with exponentially moving average (EMA) of student model parameters $\boldsymbol{\theta}^S$. Normally in a teacher-student framework, all teacher model parameters move similarly toward the student parameters with a single smoothing parameter (momentum). However, in Tables 1 and 3, we show that a single smoothing parameter is insufficient and yields poor performance. Indeed, in our case, most of the student model parameters remain frozen, and only a small portion is updated. We propose that the teacher model's parameters corresponding to the student-frozen parameters should move at a different pace than those selected for updates. Therefore we use dual smoothing parameters (referred to as momentum parameters) based on the affine transformation of the binary mask $\mathbf{m}$ to adapt the teacher parameters $\boldsymbol{\theta}^T$.

### WEIGHTED EXPONENTIAL SMOOTHING WITH DUAL MOMENTUM

After each gradient update step ($i$) for $\mathcal{M}_S$, parameters of $\mathcal{M}_T$ are updated by EMA of the student model parameters. Typically, EMA is governed by

$$\boldsymbol{\theta}_{i+1}^T = \delta \boldsymbol{\theta}_i^T + (1 - \delta) \boldsymbol{\theta}_{i+1}^S \tag{3}$$

where $\delta$ is the smoothing parameter. Further, it has been shown in (Tarvainen & Valpola (2017); Oquab et al. (2023); Koh et al. (2022)) that setting $\delta$ to a high value (eg 0.998) maintains a stable teacher model that can be considered as a strong reference for past tasks $\{0, \dots, t-1\}$. But updating the teacher model with a single smoothing parameter in cases where parameters are masked creates dissonance and increases forgetting because all the parameters are updated with equal importance, disregarding those parameters which are selected by the gradient scoring function (where $[\mathbf{m}_{ij} = 1]$). To account for masking, we modify Eq 3 as

$$\boldsymbol{\theta}_{i+1}^T = \boldsymbol{p} \boldsymbol{\theta}_i^T + \boldsymbol{q} \boldsymbol{\theta}_{i+1}^S \tag{4}$$

where $\boldsymbol{p}$ and $\boldsymbol{q}$ denote the smoothing parameters for the teacher and student model respectively and can be computed as

$$\begin{aligned} \boldsymbol{p} &= (\gamma - \delta)\mathbf{m} + \delta \\ \boldsymbol{q} &= (\delta - \gamma)\mathbf{m} + 1 - \delta \end{aligned} \tag{5}$$

where $\gamma < \delta$. This means that the selected parameters of the teacher model ($[\mathbf{m}_{ij} = 1]$) move slightly faster towards the student model as compared to the frozen candidate parameters (where $[\mathbf{m}_{ij} = 0]$). As such, parameters where $[\mathbf{m}_{ij} = 0]$ will move at a slow rate of $\delta$, and unmasked parameters would be updated with $\gamma$. When $\gamma = \delta$, the weighted scheme becomes EMA with a single smoothing parameter. A detailed proof is given in appendix A.1.

### PHASE 2: UNSUPERVISED TEST TIME LEARNING (TTL)

After supervised training is completed, both $\mathcal{M}_T$ and $\mathcal{M}_S$ are deployed for Test Time Learning (TTL). We consider teacher ($\mathcal{M}_T$) and student ($\mathcal{M}_S$) models as two experts on different data distributions, the $\mathcal{M}_S$ on the most recent and the $\mathcal{M}_T$ on previous sessions distributions.

We take inspiration from Out Of Distribution (ODD) literature Hendrycks & Gimpel (2016), where a sample has to be identified as In Distribution (ID) for a given predictor with a score function predicting high values for ID samples as opposed to OOD samples. Recently it has been shown that using the un-normalized maximum logit output of a given predictor as an ID score is significantly more robust than softmax probability Hendrycks et al. (2019). Indeed the softmax probability is

shown to provide high probability predictions even for unknown samples Yang et al. (2021), which we want to avoid in our case. Note that for CLIP, the logit corresponds to the cosine similarity of the image batch with given text features.

Following Hendrycks et al. (2019), we use the maximum logit value of each expert as an ID score and select for each test sample the expert with the highest ID score, indicating that the sample is likely to be better represented by said expert. We then accept the pseudo label of the selected expert. Formally, the pseudo label can be calculated as follows:

$$\hat{y} = \begin{cases} \hat{y_T} & \text{if } l_T \geqslant l_S \\ \hat{y_S} & \text{otherwise} \end{cases} \tag{6}$$

where $\hat{y}$ is the accepted pseudo label and $l_T = \max(\mathcal{M}_T(\mathbf{x}))$ and $l_S = \max(\mathcal{M}_S(\mathbf{x}))$ are the maximum logit score for teacher and student model respectively, and similarly $\hat{y_T} = \arg\max(\mathcal{M}_T(\mathbf{x}))$ and $\hat{y_S} = \arg\max(\mathcal{M}_S(\mathbf{x}))$ are the pseudo labels by teacher and student models respectively. During test-time training, the student model $\mathcal{M}_S$ is updated by minimizing CLIP contrastive loss given pseudo label $\hat{y}$. In realistic settings, multiple iterations on test data are often not always possible, for example, in a streaming data pipeline. We too mimic this setting, where the entire data is processed only once during the TTL phase.

Similar to the above-mentioned supervised phase, we also here apply sparse local updates to $\mathcal{M}_S$. However, the estimation of masks based on the online data might be noisy and largely reduce the efficiency as gradients of all parameters must be estimated for each mini-batch of test samples. To overcome this, and following the assumption that test data are drawn from the distributions of all previous tasks, we leverage the masks estimated for previous tasks. We accumulate a union of the binary masks $(\mathbf{m}_u)$ over all the previously seen tasks $t$ such that $\mathbf{m}_u = \mathbf{m}_1 \cup \mathbf{m}_2 \cup \ldots \mathbf{m}_t$. To maintain the same sparsity level ($c = 0.1$) of performed updates, we further select the same top-K (K=$c$) most relevant parameters from these new masked $\mathbf{m}_u$

| Momentum ($\gamma, \lambda$) | Aircraft | | |
| | Acc. ($\uparrow$) | F. ($\downarrow$) | FTA. ($\uparrow$) |
| --- | --- | --- | --- |
| 0.9999, 0.9999 | 23.99 | 18.36 | 12.15 |
| 0.5, 0.9 | 38.41 | 3.27 | 37.64 |
| 0.7, 0.9 | 37.22 | 3.05 | 37.72 |
| 0.8, 0.9* | **39.40** | **2.61** | **38.13** |
| 0.8, 0.6 | 37.06 | 5.12 | 29.63 |
| 0.8, 0.5 | 32.95 | 3.40 | 26.33 |

Table 1: Effect of Momentum ($\gamma, \lambda$) on Average Accuracy (Acc in % ), Average Forgetting (F.) and First Task Accuracy (FTA.) *0.9999, 0.8, 0.9 have been used in the main results.

parameters based on their previously computed gradient scores.

Finally, $\mathcal{M}_T(\boldsymbol{\theta}^T)$ is updated using the same dual momentum scheme, but with different smoothing vectors $\boldsymbol{p}', \boldsymbol{q}'$ as:

$$\boldsymbol{\theta}_{i+1}^T = \boldsymbol{p}'\boldsymbol{\theta}_i^T + \boldsymbol{q}'\boldsymbol{\theta}_{i+1}^S \tag{7}$$

where $\boldsymbol{p}' = (\lambda - \delta)\mathbf{m} + \delta$ and $\boldsymbol{q}' = (\delta - \lambda)\mathbf{m} + 1 - \delta$. In the TTL phase, the momentum parameter $\lambda$ is kept such that $\gamma < \lambda < \delta$. This means that $\boldsymbol{\theta}^T$ moves more slowly in the direction of $\boldsymbol{\theta}^S$ during the TTL phase as compared to the supervised phase. As we encounter frequent and possibly noisy online updates, stability is better ensured by a slower pace of movements toward student parameters. We show the sensitivity of our method on the choice of momentum values $\lambda, \delta$ in Table 1. A high $\delta$ has been chosen to keep the Teacher model stable as shown in Tarvainen & Valpola (2017); Oquab et al. (2023); Koh et al. (2022). It can be seen that when $\gamma = \lambda$ (single momentum EMA), the performance significantly drops. DoSAPP is less sensitive to on choice of $\gamma$, but it highly depends on $\lambda$. We can also see that as $\lambda < \gamma$, the performance again drops. The algorithm can be fully understood as given in algorithm block 1.

## 4 EXPERIMENTS

### 4.1 SETUP

**Architecture:** We apply DoSAPP to vision-language classification tasks, given their relatively robust knowledge measurement in such tasks. CLIP-ViT-B/16 Radford et al. (2021), is used as backbone. We report the accuracies recorded by the Teacher model. We refer to Zhang et al. (2023b) for hyperparameters selection other than dual momentums, which are given in Appendix A.2.

| Method | Aircraft Acc. (↑) | F. (↓) | Cars Acc. (↑) | F. (↓) | CIFAR100 Acc. (↑) | F. (↓) | CUB Acc. (↑) | F. (↓) | GTSRB Acc. (↑) | F. (↓) |
|---|---|---|---|---|---|---|---|---|---|---|
| CLIP-Zeroshot Radford et al. (2021) | 24.45 | - | 64.63 | - | 68.25 | - | 55.13 | - | 43.38 | - |
| Finetune Goyal et al. (2023) | 18.63 | 39.93 | 51.64 | 25.65 | 46.26 | 37.78 | 45.74 | 26.62 | 21.76 | 55.48 |
| Self-Labelling | 10.81 | 50.81 | 23.49 | 30.42 | 38.03 | 42.67 | 28.60 | 33.82 | 5.14 | 62.31 |
| MAS Aljundi et al. (2018) | 33.69 | 27.50 | 69.43 | 9.18 | 63.88 | 21.16 | 61.72 | 12.05 | 42.04 | 25.38 |
| L2P Wang et al. (2022e) | 32.20 | 21.73 | 67.04 | 11.22 | 67.71 | 18.81 | 64.04 | 6.82 | **75.45** | 2.68 |
| DualPrompt Wang et al. (2022d) | 26.61 | 17.20 | 63.30 | 18.67 | 61.72 | 19.87 | 64.38 | 12.94 | 69.65 | 8.43 |
| SLCA Zhang et al. (2023a) | 29.40 | **11.45** | 62.65 | 4.42 | 70.03 | **0.19** | 53.87 | 7.75 | 46.01 | **0.83** |
| ZSCL Zheng et al. (2023) | 30.96 | 15.65 | 67.79 | 8.27 | **80.50** | 1.05 | 61.09 | 7.69 | 62.92 | 13.54 |
| SparseCL Wang et al. (2022c) | 31.95 | 19.77 | 71.57 | 5.38 | 69.35 | 15.23 | 62.50 | 9.66 | 48.99 | 24.91 |
| SPU Zhang et al. (2023b) | 30.94 | 28.36 | 69.41 | 16.91 | 58.80 | 26.37 | 62.31 | 7.2 | 43.06 | 19.16 |
| DoSAPP | **39.14** | 12.55 | **74.87** | **-0.74** | 79.16 | 7.73 | **68.17** | 2.15 | 72.33 | 1.02 |
|  | ±0.73 | ±0.22 | ±0.03 | ±0.68 | ±0.42 | ±1.68 | ±1.24 | ±0.81 | ±0.89 | ±2.10 |
| ER methods(ER=1000) |  |  |  |  |  |  |  |  |  |  |
| ER French (1999) | 41.42 | 31.38 | 69.08 | 16.42 | 82.86 | 3.41 | 64.07 | 17.72 | **96.28** | -7.48 |
| ER + LWF Li & Hoiem (2017) | 36.08 | 18.12 | 72.56 | 4.04 | 74.32 | 8.16 | 65.11 | 5.90 | 53.56 | 11.86 |
| ER + PRD Asadi et al. (2023) | 37.11 | 17.35 | 74.08 | 3.75 | 79.66 | 3.10 | 65.92 | 6.55 | 63.00 | 12.44 |
| SPU + ER | 44.43 | 14.42 | 77.51 | **3.26** | 83.99 | -0.39 | 71.51 | 4.84 | 94.25 | **-7.87** |
| DoSAPP + ER=200 | **47.32** | **8.10** | **79.17** | 3.92 | **88.41** | **-1.96** | **74.39** | 2.77 | 83.67 | 1.92 |
|  | ±0.84 | ± 0.79 | ± 1.02 | 0.63 | ± 1.01 | ±0.09 | ±0.91 | ±0.58 | ± 0.95 | ± 0.28 |

Table 2: Acc. (Average Accuracy, ↑) and F. (Forgetting, ↓) of different methods all using CLIP ViT-B/16 backbone with trainable vision and text encoders, without any Replay Buffer in CIL scenario. DoSAPP can achieve positive backward transfer - forgetting is negative on Cars data. All experiments are mean of 5 experiments with random seeds. STD. is not shown for baselines for the ease of reading and space constraints.

**Datasets:** We consider five different vision datasets, three fine-grained (*Aircraft* Maji et al. (2013), *CUB* Wah et al. (2011), *Stanford Cars* Krause et al. (2013), *Oxford Pets* Parkhi et al. (2012), one coarse dataset (*CIFAR100* Krizhevsky (2012)) and one out-of-distribution dataset (*GSTRB* Stallkamp et al. (2012)). These datasets are chosen primarily based on their initially low zero-shot performance with CLIP pre-trained models. To form the continual learning sequences, we split each dataset into 10 subsets with disjoint classes composing 10 tasks. For all the datasets, the training data is used in the supervised learning phase. The test data is divided into 2 splits, namely $\mathcal{D}^u, \mathcal{D}^e$ where $\mathcal{D}^u$ is utilized for test-time unsupervised learning and $\mathcal{D}^e$ is used for evaluation.

**Evaluation Metrics:** After each supervised session $t_i$ and the following test-time adaptation session, we evaluate the model test performance on holdout datasets from all $T$ tasks. To do this, we construct the matrix $R \in \mathbb{R}^{T \times T}$, where $R_{i,j}$ is the test classification accuracy of the model on task $t_j$ after observing the last sample from task $t_i$. Thus, we compute **Average Accuracy** (Acc. $= \frac{1}{T} \sum_{i=1}^{T} R_{T,i}$.) and **Average Forgetting** (F. $= -\frac{1}{T-1} \sum_{i=1}^{T-1} R_{T,i} - R_{i,i}$.) Lopez-Paz & Ranzato (2017). Taken together, these two metrics allow us to assess how well a continual learner solves a classification problem while overcoming forgetting. All experiments have been done on NVIDIA A100 GPU and each one takes approximately 1 hour for completion.

## 4.2 RESULTS

We compare a variety of baselines with our proposed method in Table. 2, in the challenging scenario of class incremental learning (CIL). Along with the methods mentioned in Table. 2, we also compare our method with self-labeling (SL), where the groundtruth pseudo label comes from the trained model itself (without any student-teacher framework). When comparing methods without ER, DoSAPP achieves state-of-the-art results in all the five datasets used in the experiments. This highlights the fact that test time data can be utilized for improving transferability as well as preserving previously learned knowledge. Even when comparing methods with ER, DoSAPP (without ER) gives a comparable performance in almost all the datasets. We note that SPU+ER employs a very high buffer of 1000, which is attributed to such a high performance in some datasets like Cifar100 and GTSRB. Although our method is robust enough to be used without ER and our primary motivation is to circumvent the usage of buffer, we still present results with a small buffer (DoSAPP+ER, ER=200), for a comparison to the baselines using ER. DoSAPP + ER outperforms all other baselines except GTSRB by a significant margin.

| ID | Description | Aircraft | | Cars | | CIFAR100 | | CUB | | GTSRB | |
|----|-------------|----------|-----|------|-----|----------|-----|-----|-----|-------|-----|
| | | Acc. (↑) | F. (↓) | Acc. (↑) | F. (↓) | Acc. (↑) | F. (↓) | Acc. (↑) | F. (↓) | Acc. (↑) | F. (↓) |
| A1 | Only Teacher-Student | 30.12 | 3.50 | 67.72 | 3.66 | 77.82 | **5.17** | 62.67 | 4.11 | 53.57 | 5.38 |
| A2 | A1 + sparse params | 34.16 | 8.61 | 69.42 | 3.41 | 71.93 | 8.24 | 66.32 | 3.98 | 55.32 | 5.81 |
| A3 | A2 + EMA($\delta$) + $\mathbf{m}_u$ | 31.79 | 10.42 | 70.99 | 3.64 | 72.66 | 8.86 | 66.98 | 3.17 | 61.54 | 4.01 |
| **A4\*** | **A2 + EMA($\delta, \gamma$) + $\mathbf{m}_u$** | **39.14** | **2.55** | **74.87** | **-0.74** | **79.16** | 7.73 | **68.17** | **2.15** | **72.33** | **1.02** |
| A5 | A4 + imbalanced TTL | 35.99 | 5.22 | 72.68 | 6.38 | 75.70 | 9.81 | 64.84 | 3.73 | 68.17 | 5.63 |

Table 3: Acc. (Average Accuracy, ↑) and F. (Forgetting, ↓) when different components of DoSAPP are incrementally added to the Student-Teacher framework referred as A1. A2 denotes the sparse parameter selection added to A1. EMA($\delta$) represents single momentum updates, while EMA($\delta, \gamma$) refers to dual momentum updates. $\mathbf{m}_u$ denotes the union of mask technique described in section 3.2. **A4** is the configuration used in our proposed DoSAPP algorithm.

| Method (CLIP) | Avg Acc. (↑) | FTA (↑) | CTA (↑) | F. (↓) |
|---------------|--------------|---------|---------|--------|
| Finetune (no TTL) | $35.24 \pm 0.87$ | $5.90 \pm 1.20$ | $75.44 \pm 0.52$ | $16.87 \pm 1.04$ |
| SPU | $39.62 \pm 1.62$ | $24.31 \pm 0.30$ | **$74.94 \pm 2.43$** | $7.32 \pm 0.38$ |
| **DoSAPP** | **$45.01 \pm 0.31$** | **$30.63 \pm 0.76$** | $71.13 \pm 1.17$ | **$2.34 \pm 0.75$** |

Table 4: Average Accuracy (Avg Acc.), First Task Accuracy (FTA), Current Task Accuracy (CTA), and Average Forgetting (F.) measured for a long sequence of tasks from the concatenation of the Aircraft Maji et al. (2013) and Cars Krause et al. (2013) datasets. All results are the mean of 5 randomized experiments with different seeds.

### 4.3 Class Incremental Long Sequence scenario with domain shift

We also consider the case where we have a long sequence of tasks, each to be trained in a class incremental fashion. For these experiments, we combined the 10 tasks of Aircraft data Maji et al. (2013) and 10 tasks of Cars data Krause et al. (2013). This firstly creates a long sequence of tasks in a class incremental scenario, and secondly causes a domain shift after 10 tasks of aircraft. From Table 4, it can be clearly seen that our proposed method, DoSAPP, outperforms SPU without ER and Finetune (without any TTL phase). Further, it can be inferred that in other baselines, there is a recency bias towards the current task, whereas in DoSAPP, with a marginal decrease of 3.8% on current task accuracy (CTA), there is an overall increase in the average accuracy and the first task accuracy. This shows that our approach retains the knowledge on the first task as well as adapts to the current task, with strong generalization performance.

## 5 Ablation Study

In this section, we quantitatively analyze the effect of different components of our proposed method DoSAPP. We evaluate the effects of each component incrementally, as seen in Table 3. Starting with only a student and teacher model setup, we subject it to TTL data and this forms our baseline. Next, we compare with localized sparse updates for the first MLP layer of each of the transformer blocks. This gives an increase in performance in 4 out of 5 datasets. It is to be noted that the momentum used to update the teacher model is according to Eq. 3. We then take the union of supervised task masks to use them at the TTL phase, but this deteriorates performance since the masked parameters and unmasked parameters are updated with a single momentum. Finally, we add our dual momentum approach, which gives the best performance. We also subject our approach to a more challenging scenario where the tasks in TTL phases are class-imbalanced. Here we sample each task from a symmetric Dirichlet distribution whose concentration parameter is the length of each task. This causes a high imbalance of classes within each task, and sometimes, even absence of certain classes. This imbalanced case is of particular importance since, in real settings, test suites are often skewed. This is done by randomly sampling classes from a Dirichlet distribution. Although the performance is inferior to the balanced case, it should not be interpreted as a drawback. This is because the model should adapt more to the classes that are seen often in TTL phases and loss of performance on rarely seen classes is but natural.

| Method | Aircraft | | Cars | | CIFAR100 | | CUB | | GTSRB | |
|--------|----------|---------|--------|---------|----------|---------|---------|---------|---------|---------|
| | Acc. ($\uparrow$) | F. ($\downarrow$) | Acc. ($\uparrow$) | F. ($\downarrow$) | Acc. ($\uparrow$) | F. ($\downarrow$) | Acc. ($\uparrow$) | F. ($\downarrow$) | Acc. ($\uparrow$) | F. ($\downarrow$) |
| SPU | 30.94 | 28.36 | 69.41 | 16.91 | 58.80 | 26.37 | 62.31 | 7.2 | 43.06 | 19.16 |
| SPU+$D^u$ | 27.72 | 24.86 | 68.91 | 7.34 | 74.09 | 10.43 | 61.21 | 4.01 | 60.17 | 6.94 |
| SparsCL+RMT | 27.11 | 16.29 | 69.81 | 17.22 | 70.82 | 12.25 | 60.03 | 10.58 | 51.98 | 11.40 |
| SPU+RMT | 29.33 | 15.10 | 62.32 | 21.95 | 63.06 | 23.28 | 63.87 | 6.34 | 54.13 | 17.56 |
| **DoSAPP** | **39.14** | **12.55** | **74.87** | **-0.74** | **79.16** | **7.73** | **68.17** | **2.15** | **72.33** | **1.02** |

Table 5: Acc. (Average Accuracy, $\uparrow$) and F. (Forgetting, $\downarrow$) for comparing CIL methods like SPU Zhang et al. (2023b), SparsCL Wang et al. (2022c) integrated with most recent TTA method: RMT Döbler et al. (2023) with our proposed method: DoSAPP. It can be observed that typically fusing typical TTA method in CIL pipeline exacerbates the catastrophic forgetting. DoSAPP on the other hand outperforms all of them, by a significant margin on all the datasets.

We highlight the innovative aspect of our approach, which leverages unsupervised test data—readily available in production environments, to enhance continual learning. Unlike our method, existing continual learning (CL) techniques are not inherently designed to incorporate unsupervised test data, making them less adaptable to this scenario. Indeed, naive approaches to using the unsupervised data alongside existing methods proved unfruitful in our preliminary analysis. To illustrate this, we combined the best-performing CL method (compared to ours), SPU, with a simple pseudo-labeling baseline, namely SPU + test-time data ($D^u$). The model is updated with SPU-learned masks using a standard self-labeling approach on test-time data, using the max logit of the model as the label. Further, we integrate RMT Döbler et al. (2023), one of the most recent Test Time Adaptation methods, with SPU Zhang et al. (2023b) and SparsCL Wang et al. (2022c), and observed that our proposed method DoSAPP outperforms all of them as shown in Table 5. This highlights that TTA methods, when fused with continuous supervised training pipeline, cause the model to significantly lose knowledge. As there are long sequences of distinct tasks, it becomes difficult for any TTA method to adapt to these ever-changing source distributions. Our method mitigates this issue by intuitive usage of dual momentum over masked parameters. We further observe that the TTA method gives inferior performance for almost all datasets in comparison to self-labeling, proving that these methods are not suitable for deploying under continuous supervised learning and expanding tasks. Further in Appendx A.3 and A.4, we demonstrate the superiority of our method in adapting to noise present in the unsupervised test data and the effect of the proportion of test data $D^u$ on the performance of DoSAPP.

## 6 LIMITATION

DoSAPP is a robust algorithm which can be potentially applied to any CL technique for unsupervised adaptation of Test Time Data. However, since it utilizes the test data, its primary bottleneck becomes the quality of test data, especially if it's highly skewed. Another limitation is the increase in the computational budget due to two deployed models: Student-Teacher framework. We address this by leveraging the efficient sparse parameter selection method.

## 7 DISCUSSION AND CONCLUSION

In this work, we discuss how to leverage test-time data to improve models' representation of previous tasks, mimicking human learning and striving for real intelligent agents. In summary, to the best of our knowledge, we are the first to explore test-time learning to control forgetting. We show that test-time data can provide a great source of information when leveraged correctly. Our method, DoSAPP, was able to significantly improve over the zero-shot performance of CLIP when continually learning a dataset without any replay and with no specific CL method applied at the supervised training session. DoSAPP is stable due to sparse parameter updates and the weighted EMA teacher-student framework. Further, during TTL, the max-logit in distribution scores makes it more robust to class imbalance than other strategies.

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

# A    APPENDIX / SUPPLEMENTAL MATERIAL

## A.1    DERIVATION FOR DUAL MOMENTUM

In section 3, the teacher model parameters $\boldsymbol{\theta}_i^T$ undergo exponential moving average as

$$\boldsymbol{\theta}_{i+1}^T = \boldsymbol{p}\boldsymbol{\theta}_i^T + \boldsymbol{q}\boldsymbol{\theta}_{i+1}^S \tag{8}$$

where $\boldsymbol{p}$ and $\boldsymbol{q}$ denote the smoothing parameters for the teacher and student model respectively and can be computed as

$$\begin{aligned} \boldsymbol{p} &= \alpha_1 \mathbf{m} + \beta_1 \\ \boldsymbol{p} &= \alpha_2 \mathbf{m} + \beta_2 \end{aligned} \tag{9}$$

where $\alpha_i$ and $\beta_i$ for $i \in \{1, 2\}$ are the coefficients for the affine transformation of the boolean mask vector $\mathbf{m}$.

To account for masked parameters, two momentum values $\delta, \gamma$ are introduced for teacher and student models respectively, such that for the teacher model, affine coefficients $\alpha_1, \ \beta_1$ are computed by solving the equations:

$$\alpha_1 [\mathbf{m}_{ij} = 1] + \beta_1 = \gamma \ , \qquad \alpha_1 [\mathbf{m}_{ij} = 0] + \beta_1 = \delta \tag{10}$$

and $\alpha_2, \beta_2$ are computed by solving the equations

$$\alpha_2 [\mathbf{m}_{ij} = 1] + \beta_2 = 1 - \gamma \ , \qquad \alpha_2 [\mathbf{m}_{ij} = 0] + \beta_2 = 1 - \delta \tag{11}$$

This gives

$$\begin{aligned} \alpha_1 &= \gamma - \delta, \quad \beta_1 = \delta \\ \alpha_2 &= \delta - \gamma, \quad \beta_2 = 1 - \delta \end{aligned} \tag{12}$$

This gives

$$\begin{aligned} \boldsymbol{p} &= (\gamma - \delta)\mathbf{m} + \delta \\ \boldsymbol{q} &= (\delta - \gamma)\mathbf{m} + 1 - \delta \end{aligned} \tag{13}$$

## A.2    HYPERPARAMETERS

Table 6 shows different hyperparameters that have been used for all the experiments using CLIP backbones. The hyperparameters were selected based on the performance of the first task of Cars Krause et al. (2013) dataset. All the results have been gathered over experiments averaged over 5 random seeds.

| Hparams | CLIP model |
|---|---|
| Batch Size | 64 |
| Optimizer | AdamW |
| Learning Rate | $7.5e - 6$ |
| CL Epochs | 10 |
| Buffer | 0 |
| TTL batch size | 64 |
| Momentum-EMA ($\delta, \gamma, \lambda$) | 0.9999, 0.8, 0.9 |
| sparsity ($c$) | 0.1 |

Table 6: Hyper Parameters for all the experiments using CLIP ViT-B/16 model.

## A.3    DEPENDENCE ON QUALITY OF TEST DATA USED FOR UNSUPERVISED LEARNING

We want to highlight that the trained model is expected to generalize to the distribution of the test data. We also assume that any quality degradation will be consistent across time steps. For instance, if

the data is corrupted with noise, our method would generalize and adapt the model to this corruption as well. To illustrate this, we conducted a small experiment by adding random Gaussian noise (mean = 0, std = 0.1) to different combinations of the test and evaluation suite (referred to as GN in Table 7). The results are shown below, with average accuracy (Acc.) followed by forgetting (F.). We observe that when corruption is present in the test-time data, the model is still able to leverage these data and improve on clean evaluation data compared to the test-time baseline by a significant margin of 17% (SPU alone). Interestingly, the model adapted to test-time data with Gaussian noise performs better on evaluation data with Gausian noise than the case when the test-time data is clean. This is the evidence of our method's ability to adapt and generalize to the present test-time conditions.

| Test Time Data ($D^u$) | Evaluation Data ($D^e$) | Acc. ($\uparrow$) | F. ($\downarrow$) |
|---|---|---|---|
| Clean | Clean | 79.16 | 7.73 |
| GN | Clean | 75.67 | 9.93 |
| Clean | GN | 69.50 | 12.86 |
| GN | GN | 73.42 | 6.86 |

Table 7: Performance of DoSAPP with noise added to $D^u$ and $D^e$ for CIFAR100 Data

### A.4 ABLATION STUDY ABOUT THE SIZE OF TEST-TIME DATA $D^u$

In our method, we divided the evaluation data into two halves. One half is for unsupervised learning ($D^u$), and the other half is for evaluation ($D^e$). In the table below, we feed the fraction of $D^u$ for test time learning. 0.25 means that 25% of the original $D^u$ is fed to the model for unsupervised learning. We notice that when the fraction is below 0.75, there is an appreciable difference between the performance of our proposed model. However, at 0.75, the performance is quite close to that of the whole $D^u$.

| Fraction of $D^u$ | Acc. ($\uparrow$) | F. ($\downarrow$) |
|---|---|---|
| 0.25 | 73.97 | 14.23 |
| 0.5 | 76.83 | 9.44 |
| 0.75 | 79.02 | 8.16 |
| 1 | 79.16 | 7.73 |

Table 8: Dependence of performance of DoSAPP with different proportion of the testing data $D^u$ on CIFAR100 dataset.

