# OpenReview forum: "Controlling Forgetting with Test-Time Data in Continual Learning"
_ICLR.cc/2025/Conference — Submitted to ICLR 2025_

### Official Review · Reviewer_PpH5 · 2024-10-28

**Soundness:** 3
**Presentation:** 3
**Contribution:** 3
**Rating:** 5
**Confidence:** 4

**Summary:**

The authors propose to leverage test time adaptation to improve the performance of foundational models when fine-tuned continually. They show significant improvements on many benchmarks when using their proposed TTA method.

**Strengths:**

- Explores the intersection of test-time adaptation and continual learning which is an interesting direction
- Performs an ablation study
- Comparison with many existing methods

**Weaknesses:**

- **W1**: More explanation around the exact setup used for test time adaptation needs to be made. For instance, it is not clear if Test time adaptation is done image-wise, in which case it would be fair to compare a test time adaptation method to another method that does not use test time adaptation, because both methods operate in the same setting. However, if this is not the case (for instance if test time adaptation is performed on batches that come from a single task, then evaluated on that task), this is not fair at all for other methods and any comparison with non-TTA method is doomed.
- **W2:** **Lack of baselines** Related to the previous weakness, in some TTA setting (that make sense in the wild), a lot of these CL methods you compare to could benefit from a simple change that would make them able to do TTA in some way (way less complicated than the method you propose). For instance, simply by allowing them to gather batch statistics in the BN layers if any, or by inferring the current task through retaining task prototypes and then only looking at the output of this tasks head.

I cannot review the paper further without having a clear answer and explanation inside the paper on how this TTA is performed, so I invite the authors to consider explaining this part very carefully. In particular, it should be made clear **in what order** the samples come to the model during the TTA phase, and when is the evaluation perfomed ? (after all samples have been processed or immediately after processing one sample)

**Questions:**

- **Q1**: Is test time adaptation performed image-wise or batch-wise ?
- **Q2**: If it is performed batch wise, is it performed using the shuffled test set comprising all tasks at the same time, or is it performed task-by-task ?

---

> ### Author Response · Authors · 2024-11-23
> **Response to PpH5**
>
> Dear Reviewer PpH5, thank you so much for the review and the constructive suggestions! Here is our response to your questions and concerns.
>
> # Clarification around the exact setup and fair comparison with other baselines.
> 1. Our proposed method addresses the problem of forgetting in Class Incremental Learning (CIL) by incorporating unsupervised learning during the testing phase, a unique approach distinct from Test-Time Adaptation (TTA). In a production setting, where models must adapt as new classes are introduced, test data from previously learned tasks is often abundant but underutilized, serving only for evaluation. We leverage this test data with our method, DoSAPP, to mitigate forgetting through unsupervised learning. While DoSAPP can adapt to test distribution conditions, its primary goal is to reduce forgetting without requiring extensive data storage or replay during supervised training. Unlike TTA, which focuses on addressing input distribution shifts (e.g., data corruption), our method assumes the test data distribution aligns with the training data of learned tasks, enabling effective continual learning.
>
> 2. In conventional TTA methods,  both adaptation and evaluation are performed on the same test-time data. In contrast, our approach partitions the test data into two distinct subsets: one dedicated to unsupervised learning(Du) and the other reserved for evaluation($D^e$). This partitioning is merely to assess the generalization of the model ensuring a fair assessment of the model’s performance.
>
> 3. The comparison between our method and other methods is fair since other methods are also trying to improve CIL performance. We agree that other methods do not utilize test data, however, this is part of the novelty of our approach. In Table 5 in the revised rebuttal submission, we present results combining SPU and SparsCL with vanilla pseudo labeling, along with integrating the most recent TTA method, RMT[a]. We demonstrate that combining typical TTA with CIL gives inferior performance. We further emphasize that typical TTA methods such as test time normalization do not apply directly here as they are designed for input distribution shifts such as corruptions which do not impact the label distributions[b].
>
> [a] Robust Mean Teacher for Continual and Gradual Test-Time Adaptation by Mario et al(CVPR 2023)
>
> [b] Channel-Selective Normalization for Label-Shift Robust Test-Time Adaptation by Vianna  et al(CoLLAs, 2024)
>
> # Lack of baselines
> 1. With regards to using batch normalization-based methods (test time normalization), we want to also highlight that we use a CLIP ViT model which does not incorporate batch normalization, thus those approaches are not directly applicable. We added more experimental results in Table 5 by integrating RMT[a] the most recent TTA state-of-the-art method, with SPU and SparseCL. It can be observed that our proposed method DoSAPP outperforms all of them. This highlights that TTA methods when fused with a continuous supervised training pipeline cause the model to significantly lose knowledge. As there are long sequences of distinct tasks, it becomes difficult for any TTA method to adapt to these ever-changing source distributions. On the other hand, our method mitigates this issue by the unique design of EMA with dual momentum. We further observe that the TTA method gives inferior performance for almost all datasets in comparison to self-labeling, proving that typical TTA is not suitable for deployment under continuous supervised learning, and expanding tasks.
> | Method                              | Cifar100         |     Aircraft      |     Cars         | GTSRB        | Cub          |
> |---------------------------|----------------|-----------------|--------------------|------------------|--------------|
> | SPU + Pseudo Labeling| 74.09, 10.43| 27.72, 24.86  | 68.91, 7.34  | 60.17, 6.94  | 61.21, 4.01  |
> | RMT + SPU                           | 63.06, 23.28 | 29.33, 15.10  | 62.32, 21.95 | 54.13, 17.56 | 63.87, 6.34  |
> | RMT + SparseCL                      | 70.82, 12.25| 27.11, 16.29  | 69.81, 17.22 | 51.98, 11.40 | 60.03, 10.58 |
> | **DoSAPP (from paper)**                | **79.16, 7.73** | **39.14, 12.55**  | **74.87, -0.74** | **72.33, 1.02**  | **68.17, 2.15** |
>
> **Table-A** Accuracy and Forgetting for CIL methods integrated with RMT compared with DoSAPP over different datasets.
>
> [a] Robust Mean Teacher for Continual and Gradual Test-Time Adaptation by Döbler et al(CVPR 2023)
>
> # Addressing Questions:
> Q-1: Is test time adaptation performed image-wise or batch-wise ?
>
> A-1: Batch wise on $D^u$
>
>
> Q2: If it is performed batch wise, is it performed using the shuffled test set comprising all tasks at the same time, or is it performed task-by-task ?
>
> A-2 Yes, its batchwise on a shuffled test set $D^u$ comprising of all the previous tasks seen so far, and evaluation is done on a separate partition($D^e$). This can be referred in the Fig 1 and the algorithm block.

---

> > ### Author Response · Authors · 2024-11-25
> > **Follow up to PpH5**
> >
> > As the deadline for rebuttal is coming closer we are following up regarding our response to your review. We understand you may have a busy schedule, and we truly appreciate the time and effort you dedicate to providing valuable feedback.
> >
> > We wanted to kindly check if you’ve had an opportunity to review our response.
> >
> > If there’s any additional information or clarification we can offer, please feel free to let us know.
> >
> > We thank you once again for your time. We look forward to your response.

---

> > ### Comment · Reviewer_PpH5 · 2024-11-26
> >
> > Thank you for the answers and the changes to the manuscript. In particular the fact that test time adaptation is performed on the shuffled test set comprising all tasks seen so far removed my doubts about the method being able to infer the task using test time information, which would be unfair compared to other methods. I also like that you added a comparison to combination of CL methods and TTA method. I will raise my rating accordingly.
> >
> > However, what I don't understand then is what is so special with the link between continual learning and TTA if other CL methods do not benefit from application of TTA. Is your method good only in combination with CL or is it also a good TTA method (outside of CL context) ? I don't like overall the approach of the paper that proposes first a method before investigating the use of TTA in CL (which I don't think a lot of papers did before so it's a necessary step), and the method proposed looks highly over-engineered which might pose robustness problems.

---

> > > ### Author Response · Authors · 2024-11-27
> > > **Response to PpH5**
> > >
> > > We sincerely thank you for your thoughtful review and for increasing the score accordingly. We appreciate the opportunity to clarify your further concerns.
> > >
> > > In TTA literature, existing methods typically assume a pretrained model that adapts to input distribution shift(covariate). These methods however do not address the challenges of continual learning(CL), especially under label shift. Even methods like CoTTA[a] focus solely on handling continual shifts in the test stream and are not designed for the continual learning paradigm, which entails adapting to sequentially received supervised data under label shift. Our work is focused on addressing this latter problem.
> > >
> > > To address your concerns:
> > > 1.  **Presentation:** We would like to highlight that our paper first motivates and proposes the setting and then one solution. In the introduction the setting is motivated and proposed, our first contribution (Line 107) is explicitly the setting and not the method, and Figure 1 demonstrates the setting without any reference to a solution. Subsequently, we highlight our method which is one way to address this setting which we have found to be the best performing. Note that similarly in the Methodology setting, first the problem setting is described (Sec 3.1), and then a method is described (Sec 3.2). To further address the reviewer's concern we have now added a small paragraph at the start of section 3(Lines 188-194) that highlights this distinction.
> > >
> > > 2. **Distinction from TTA:** Our method is not merely a combination of CL with TTA approaches. It's a simple yet intuitive CL algorithm designed to mitigate catastrophic forgetting by using task-specific parameter masking, dual momentum updates, and logit comparisons between the teacher and student models at test time. The overarching question that we aim to answer is - _Can test data be utilized to mitigate forgetting in class incremental continual learning_? This is different from the typical TTA algorithms trying to adapt to input distribution shifts in data.
> > >
> > > * Our method is not just another variation or combination of TTA approaches. Unlike typical TTA methods that address data corruption, our approach leverages unsupervised data from previous tasks, requiring no corruption and effectively handling changes in label distributions.
> > >
> > > * While it is possible that our setting and other CL methods could benefit from methods in the TTA literature, our rebuttal experiments show that naively applying existing TTA algorithms in a CL pipeline often yields suboptimal results, sometimes causing more harm than good. We have added these newfound results in Table 5 of the revised rebuttal submission.
> > >
> > > * To address the reviewer's concern we have further revised the related work section Lines 171-180 in the revised rebuttal submission.
> > >
> > > 3. **Applicability of DosAPP in TTA (out of CL context)**:  We believe this falls outside the scope of our work. Our method relies on a set of interlinked components specifically tailored to the CL paradigm(e.g. task-specific parameter masking) These components are fundamentally designed to address the challenges of CL, such as task interference and catastrophic forgetting. While extending this approach to handle covariate shifts in a typical TTA setting would be an interesting direction, it would require a specialized loss design, which is beyond the focus of this study.
> > >
> > > 4. **Robustness:** Regarding your concern about robustness, we demonstrated in Appendix A.3 that our algorithm performs reliably even under noisy test data, which supports its resilience.
> > >
> > > 5. **“Over-engineered”:** We respectfully disagree that the method is "over-engineered." Unlike other approaches (e.g., L2P[b], DualPrompt[c]), our method does not introduce external modules and, in fact, reduces the number of parameters through masking. Compared to L2P, which utilizes three hyperparameters (prompt pool size, single prompt length, and selection window size), our method introduces just one additional hyperparameter: the dual momentum term. By strategically applying dual momentum to update masked parameters and comparing logit scores during unsupervised training, our approach remains straightforward yet effective in retaining knowledge in foundational models like CLIP. Our experiments and ablation studies highlight the importance of each component in achieving the method's effectiveness in this context.
> > >
> > >  [a] Continual Test-Time Domain Adaptation by Wang et al(CVPR 2022)
> > >
> > >  [b] Learning to Prompt for Continual Learning by Wang et al(CVPR 2022)
> > >
> > >  [c] DualPrompt: Complementary Prompting for Rehearsal-free Continual Learning by Wang et al (CVPR 2022)

---

### Official Review · Reviewer_uBk1 · 2024-10-31

**Soundness:** 2
**Presentation:** 2
**Contribution:** 3
**Rating:** 6
**Confidence:** 4

**Summary:**

The authors propose a novel approach to Continual Learning that enables the learner to benefit from unsupervised data available during the test phase, addressing a challenge known as Test-Time Adaptation (TTA). The approach can be divided into two main contributions: the first focuses on mitigating catastrophic forgetting during training on sequential tasks, while the second introduces test-time adaptation (applied during the evaluation phase). Both processes leverage the well-established teacher-student paradigm, incorporating a dual mechanism to control the rate at which each weight of the teacher moves towards the corresponding weight of the student (dual momentum). Experiments demonstrate that the proposed architecture achieves good performance.

**Strengths:**

- The setting explored by the authors is noteworthy and realistic.
- The approach is technically solid.
- The concept of dual momentum is promising, though I have a few questions for the authors (see below).

**Weaknesses:**

My primary concerns pertain to the **experimental setup**. Specifically, I found the comparison in Table 1 somewhat misleading. While DoSAPP leverages additional (albeit unsupervised) data from all tasks, the other approaches are standard continual learning methods that do not utilize test-time data. To their credit, the authors discuss this and include an additional comparison in Table 5 with a simple baseline combining SPU and pseudo-labeling. The authors justify this baseline by stating that SPU is the best-performing continual learning (CL) method; however, this claim does not hold up when examining the results in Table 2. For instance, SparseCL outperforms SPU, as does ZSCL (on average). Consequently, I feel the comparison is currently weak and should be expanded.
Additionally, while pseudo-labeling is a simple and effective approach, it only scratches the surface of Test-Time Adaptation (TTA), with many more advanced and effective methods proposed in recent years. Confidence in the technical contributions of this work (specifically, those outlined in Phase 2) would be strengthened if the authors extended their comparative analysis to include more recent TTA approaches. For example, the authors of CoTTA (Continual TTA) compared their approach not only with pseudo-labeling but also with BatchNorm adaptation and TENT (the latter outperforms pseudo-labeling).

I understand that these suggestions might sound like the typical “please, provide more experiments.” However, when a novel experimental setting bridges two or more fields, it is essential to design experiments that thoroughly consider the current state-of-the-art in each field.

Finally, I recommend that the authors correct the use of bold font in Table 2, where it should highlight top-performing methods rather than the proposed method. For instance, L2P achieves an accuracy of 75.45 on GTSRB, which is 3 points higher than DoSAPP (currently in bold). The same issue occurs with CIFAR-100 (where ZSCL performs better) and with the forgetting metric on Aircraft, CIFAR-100, CUB, and GTSRB.

**Clarity**
The concept of dual momentum is interesting, yet I believe there is a clarity issue in the paper. The authors state that a subset of parameters is frozen during training, following Zhang et al. (2023b). However, if certain student parameters remain unchanged during training, I would assume that their corresponding teacher parameters (updated via EMA) would also remain unchanged. This raises a question about how EMA could affect parameters that are kept frozen (line 311). I’m uncertain if this confusion arises from a lack of clarity in the paper or my own interpretation, but even after multiple readings, I could not resolve it.

**Presentation**

The paper is generally clear and well-written; however:
- In the section titled "Teacher Student Framework," the symbol indicating the parameter set is missing (see Eq. 3). Based on Fig. 2, it appears the symbol was \theta. I recommend the authors add it, as its absence complicates reading.
- Table 4 is presented inline within the text. This is acceptable, but it would be helpful if it appeared closer to its reference in Sec. 4.3.
- I suggest the authors refine punctuation and correct spacing between consecutive words, parentheses, citations, etc. For example, there are multiple instances where a blank space is missing between a word and a parenthesis.
- In Section A.3, the last line seems to be leftover text from a previous submission.

**Questions:**

I do not have questions for the authors.

---

> ### Author Response · Authors · 2024-11-23
> **Response to uBk1**
>
> Dear Reviewer uBk1, thank you so much for the review and the constructive suggestions! Here is our response to your questions and concerns.
>
> # Addressing Weaknesses
> ## Combining TTA methods with other baselines.
>
> We added more experimental results in Table 5 by integrating RMT[a] the most recent TTA state-of-the-art method, with SPU and SparseCL. It can be observed that our proposed method DoSAPP outperforms all of them. This highlights that TTA methods when fused with continuous supervised training pipeline cause the model to significantly lose knowledge. As there are long sequences of distinct tasks, it becomes difficult for any TTA method to adapt to these ever-changing source distributions. On the other hand, our method mitigates this issue by the unique design of EMA with dual momentum. We further observe that the TTA method gives inferior performance for almost all datasets in comparison to self-labeling, proving that typical TTA is not suitable for deployment under continuous supervised learning, and expanding tasks.
>
>
> | Method                              | Cifar100         |     Aircraft      |     Cars         | GTSRB        | Cub          |
> |---------------------------|----------------|-----------------|--------------------|------------------|--------------|
> | SPU + Pseudo Labeling| 74.09, 10.43| 27.72, 24.86  | 68.91, 7.34  | 60.17, 6.94  | 61.21, 4.01  |
> | RMT + SPU                           | 63.06, 23.28 | 29.33, 15.10  | 62.32, 21.95 | 54.13, 17.56 | 63.87, 6.34  |
> | RMT + SparseCL                      | 70.82, 12.25| 27.11, 16.29  | 69.81, 17.22 | 51.98, 11.40 | 60.03, 10.58 |
> | **DoSAPP (from paper)**                | **79.16, 7.73** | **39.14, 12.55**  | **74.87, -0.74** | **72.33, 1.02**  | **68.17, 2.15** |
>
> **Table-A** Accuracy and Forgetting for CIL methods integrated with RMT compared with DoSAPP over different datasets.
>
> [a] Robust Mean Teacher for Continual and Gradual Test-Time Adaptation by Döbler et al(CVPR 2023)
>
> ## Addressing the formatting issues/typos
>
> We have corrected the bold font usage in the table for both metrics.
>
> ## Clarifying the dual momentum concept
>
> 1. In our proposed method, we update only the first MLP layer's parameters (as detailed in lines 258-262 of the revised rebuttal submission). All other parameters remain frozen throughout the training process for both the student and teacher models, ensuring they are not modified during any phase of training.
>
> 2. During the supervised phase of the current task, we first estimate the parameter's score”(given by equation 2 in the paper) based on the gradient norm for all candidate parameters in the first MLP layer of each Transformer block. Using this score, we generate a binary mask [Zhang et al., 2023b], which selects the top 10% of those candidate parameters. These selected parameters are the only ones updated during the student model's training.
>
> 3) The goal of EMA is to move the teacher smoothly and steadily towards the student model, we observed that applying the standard EMA to all the first MLP layer parameters similarly (both M=0 and M=1) or only to selected parameters (M=1) results in instability in the teacher due to parameters moving at different paces than those of the student. This is where our dual momentum mitigates the logical inconsistency of updates, and provides a smoother and robust average, as given in lines 286-289, and in the section entitled “weighted exponential smoothing with dual momentum”. We further provide an ablation to demonstrate the instability of using single momentum in the table below. It can be clearly observed that any option of single momentum creates a dissonance in training, and dual momentum as proposed in our method, gives stable and better results.
>
> | Method                                | Aircraft | Cars  | Cifar100 | Cub   | GTSRB | Average |
> |---------------------------------------|----------|-------|----------|-------|-------|---------|
> | Single Momentum applied equally (for **m**=0 and **m**=1) | 23.99    | 67.48 | 78.77    | 56.70 | 52.86 | 55.96   |
> | Single Momentum applied only for **m**=1  | 32.93    | 71.78 | 73.84    | 65.26 | 61.90 | 61.14   |
> | **Dual Momentum (from paper)**            | **39.14**    | **74.87** | **79.16**    | **68.17** | **72.33** | **66.73**   |
>
> Table B: Average accuracy for different datasets comparing single momentum with our proposed method DoSAPP. Here, **m** refers to the binary mask computed through gradient norms of candidate parameters.
>
>
> 4. Keeping other parameters frozen(eg those of the transformer) does not pose any danger to the model learning since they represent generalized knowledge in a foundational model like CLIP.
>
> ## Addressing Presentation Issues
> We have addressed all the mentioned issues in the revised rebuttal submission.

---

> > ### Author Response · Authors · 2024-11-25
> > **Follow up to uBk1**
> >
> > As the deadline for rebuttal is coming closer we are following up regarding our response to your review. We understand you may have a busy schedule, and we truly appreciate the time and effort you dedicate to providing valuable feedback.
> >
> > We wanted to kindly check if you’ve had an opportunity to review our response.
> >
> > If there’s any additional information or clarification we can offer, please feel free to let us know.
> >
> > We thank you once again for your time. We look forward to your response.

---

> > > ### Comment · Reviewer_uBk1 · 2024-11-27
> > >
> > > I thank the authors for their response and the efforts they have made to improve the paper. However, I still have some reservations and concerns.
> > >
> > > ## Combining TTA Methods with Other Baselines
> > >
> > > I appreciate the new experiments provided by the authors. However, I share the concerns raised by Reviewer PpH5 regarding the interplay between continual learning (CL) and test-time adaptation (TTA), particularly in the context of the experimental comparisons.
> > >
> > > Above all, I believe exploring the use of TTA in incremental scenarios is valuable and worth pursuing. While I do not question the validity of the proposed approach, which seems effective in practice, it is challenging to disentangle the contributions of each technique and to understand their impact relative to recent works in both the CL and TTA domains.
> > >
> > > For instance, based on the latest experiments: 1) SparseCL performs worse when combined with TTA via RMT; 2) SPU deteriorates when pseudo-labeling is applied. These results suggest, as noted by the authors, that leveraging standard TTA approaches in continual learning is problematic. However, from a paper like this, I would expect a deeper investigation into the factors contributing to such underperformance in online scenarios. Clearly identifying and analyzing these limitations would significantly enhance the paper’s impact and provide readers with a better understanding of the motivation behind the proposed approach.
> > >
> > > The authors state:
> > > > “As there are long sequences of distinct tasks, it becomes difficult for any TTA method to adapt to these ever-changing source distributions.”
> > >
> > > Since TTA uses data from all tasks, I do not see why a robust TTA method would struggle to adapt to the union of training distributions. A clearer explanation is necessary, as it would help clarify why their proposed approach is so effective instead.
> > >
> > > > “Our method mitigates this issue by the unique design of EMA with dual momentum.”
> > >
> > > While the dual momentum design is appealing, it appears conceptually similar to existing EMA teacher-student approaches. It is unclear which specific technical contributions (e.g., dual momentum, selective parameter updates, mask union) drive the effectiveness of this approach in continual learning. These contributions are interdependent, making it challenging to evaluate their individual significance relative to prior work.
> > >
> > > ## Clarifying the Dual Momentum Concept
> > >
> > > I still struggle to understand the dual momentum approach fully. If some parameters are frozen (e.g., M=0), these parameters remain static during training. Consequently, their corresponding teacher model values would also remain unchanged. This raises the question: if the moving average is applied to stationary values, as it holds for frozen parameters, how does the EMA update function for these frozen parameters? Without updates to these parameters, what role does EMA play in this context?
> > >
> > > ## Presentation
> > >
> > > I recommend adjusting the placement of Table 4 to ensure it does not appear in the first paragraph of Section 5. This adjustment would enhance the paper's readability and presentation. Moreover, as suggested by Reviewer 4MGu, I strongly recommend that the authors conduct a thorough proofreading to address issues with English, punctuation, and missing spaces.

---

> ### Author Response · Authors · 2024-11-28
> **Response to uBk1**
>
> Thank you so much for acknowledging our efforts.! Here is our response to your recent questions and concerns.
>
> # Combining TTA Methods with Other Baselines
>
> We want to emphasize that although test data is used in TTA, CoTTA, and our method, we address a fundamentally different problem: **Can test data be utilized to mitigate forgetting in class-incremental continual learning?** Unlike most of the TTA literature ([a,b]), which focuses on handling input distribution shifts (e.g., image corruption), our approach targets scenarios without such shifts. Additionally, many TTA methods relying on batch-level information struggle when label shifts occur ([d, e]) and are not designed to improve results in the absence of input distribution shifts.
>
> In our work, the unsupervised test-time data does not need to be corrupted, as we address a fundamentally different problem. In our setting, **the test-time data has a different label distribution compared to the most recent supervised data**, often including classes absent from the latest supervised data. This results in a label shift, which is a key distinction that renders many TTA methods inapplicable. Since most TTA methods are designed to handle input distribution shifts and often struggle with label shifts, it is unsurprising that they fail in our scenario, where only label shifts occur without any input distribution shifts.
>
> We also note that our choice of RMT is based on finding a method that does not explicitly target corruptions (unlike, for example, batchnorm adaptation). Although these methods do improve performance in some cases, they underperform as they are simply not designed for this setting.
>
> Pseudo-labeling may seem promising and serves as our first baseline, but it is prone to errors from the main model, which may have forgotten earlier tasks. Solely relying on model predictions amplifies bias toward recent tasks, increasing label confusion. This is where the EMA framework with a teacher-student model becomes crucial. Our simple yet elegant selection of the model that is an expert on a given test data point provides the pseudo-label. Parameter selection and masking are designed to handle large pretrained models efficiently, enabling online updates while minimizing forgetting (as per SPU). Since test-time data may originate from any previous distribution, we utilize the union of masks. Additionally, masking ensures parameters are updated at varying rates. Initial experiments revealed that plain EMA momentum updates led to optimization instability, which we addressed through the introduction of dual momentum for improved stability.
>
> Regarding the interplay between the different components and which technical contributions drive the effectiveness, **in Section 5 and Table 3, we fully ablate all components of our method.** We are happy to address any specific items that are lacking in the ablations.
>
>
> We will update our paper with these clarifications.
>
> [a] Continual Test-Time Domain Adaptation by Wang et al(CVPR 2022)
>
> [b] Robust Mean Teacher for Continual and Gradual Test-Time Adaptation by Dobler et al.(CVPR 2023)
>
> [d] Channel-Selective Normalization for Label-Shift Robust Test-Time Adaptation by Vianna et al (CoLLAs 2023)
>
> [e] DELTA: DEGRADATION-FREE FULLY TEST-TIME ADAPTATION by Zhao et al. (ICLR 2023)

---

> ### Author Response · Authors · 2024-11-28
> **Clarifying the Dual Momentum Concept and Presentation**
>
> # Clarifying the Dual Momentum Concept
> 1. As pointed out in [a,b], parameters from the first MLP layer of each transformer block, referred to as **candidate parameters**, are considered for the update. All other model parameters from other modules for both the student and teacher models remain static and are excluded from discussion.
>
> 2. During the supervised training phase of the first task, assume there are 1000 candidate parameters - all from the first MLP block of each transformer layer. Gradient norms are calculated for these parameters, and with sparsity set to 10%, 100 learnable parameters are selected and masked as ($\bf{m}=1$) (set $A$), while the rest are masked as ($\bf{m}$=0). Only these 100 parameters undergo gradient updates in the student model.
>
> 3. For EMA, after the first task, under single momentum, all 1000 candidate parameters of the teacher model receive updates. However, only parameters marked (m=1) are effectively updated for the first task. Unchanged parameters (m=0) in the student model remain unchanged in the teacher model as well, only for the first task.
>
> 4. Similarly, after the second task, gradient norm calculation is repeated, resulting in a potentially different set of selected parameters (set $B$; $B\neq A$), with possible overlaps. With standard EMA there are two options:
> - **Updating All Candidate Parameters**: Updating all 1,000 candidate parameters in the teacher model fails to account for the varying importance of selected parameters, which should be updated at a faster rate compared to other candidate parameters. **Neglecting this distinction results in uniformly slow updates, causing a significant divergence between the teacher and student models.** This issue is reflected in the poor performance metrics shown in Table B of the rebuttal experiments (Row 1).
>
> - **Updating Only Selected Parameters($m=1$)**: This approach introduces **recency bias**. For example, consider a parameter $x_a$ that was selected for task 1 (A) but not for task 2 (B), i.e., $x_a \in A \text{ but } x_a \notin B$ ($x^t_a$ denotes the teacher parameter while $x^s_a$ denotes the corresponding student parameter). When the student is updated on task 1, $x^t_a$ is moving towards $x^s_a$. However,  when the supervised session of task 1 ends, $x^t_a$ faces a **sudden halt**, while other parameters start to move towards the student model e.g.,  $x^t_b$ starts to move towards the student. These **abrupt shifts** create an imbalance as $x^t_a$ is suddenly frozen while other parameters continue or start to move. This introduces a **dissonance among parameters, thus destabilizing training.** To remedy this, our dual momentum ensures that all the candidate teacher model parameters (first layer MLP of each transformer) move towards the student model as updates are performed on the student. However, selected parameters for a given task (e.g., $B$) are updated faster than the rest, ensuring their greater significance is reflected during task $B$.
>
> To recap.
> 1. **Non-selected candidate parameters also receive ema updates but at a slower rate (smaller momentum).**
> 2. **Selected parameters ($\bf{m}=1$) are updated at a slightly faster rate.**
>
> This ensures gradual and balanced updates ensuring all parameters are moving towards the student in a stable robust manner, maintaining stability and mitigating the drift caused by masking. Our experiments confirm that dual momentum stabilizes training, unlike single momentum or selective updates, leading to teacher model instability.
>
> [a] Overcoming Generic Knowledge Loss with Selective Parameter Update by Zhang et al. CVPR(2024)
>
> [b] Transformer feed-forward layers are key-value memories by Geva et al(EMNLP 2022)
>
> # Presentation:
> We have made the corrections as suggested. We would like to confirm that we have thoroughly proofread and addressed issues with English, punctuation, and missing spaces. This can be reflected in the most recent revised rebuttal submission.

---

> > ### Comment · Reviewer_uBk1 · 2024-11-29
> >
> > I would like to thank the authors for their thorough clarification on the interplay with TTA, which I believe provides valuable insight into the scope of their work.
> >
> > Regarding dual momentum, am I correct in assuming that during the second task, if a parameter is frozen (m = 0), the corresponding value in the teacher model will converge to the frozen value of the student model? If so, this appears functionally equivalent to directly "copying" the frozen parameters from the student to the teacher after the supervised phase. Moreover, since the teacher model does not play an active role during the supervised phase, the stability considerations highlighted by the authors seem less relevant in this context. Instead, it seems that the stabilization effect of dual momentum is primarily impactful during the unsupervised phase. Would the authors agree with this interpretation or clarify their position?
> >
> > Finally, I would like to acknowledge the authors' efforts during the rebuttal period, and I will raise my score by one to reflect their contributions.

---

> > > ### Author Response · Authors · 2024-11-30
> > > **Response to uBk1**
> > >
> > > We sincerely thank you for recognizing the value of our work and for raising the score. Below, we clarify the dual momentum approach.
> > >
> > > To clarify, for the second task, if a parameter is frozen (i.e., m=0), our method dictates that the corresponding value in the teacher model should **slow down** its movement toward the frozen value of the student model. Directly copying the frozen parameters from the student model to the teacher model would **instead accelerate** the movement, which is **undesirable.** Therefore, the speed of updates for frozen parameters must be **slower compared to those for selected parameters (m=1).** Further, it should be noted that the teacher's role is to maintain as much knowledge as possible of past tasks. Hence, the low momentum, in general, is to **maintain a teacher model with low rates of plasticity and higher rates of stability.** Assigning the selected student’s parameters directly to the teacher would **significantly harm that role.**
> > >
> > > The teacher model, leveraging dual momentum, plays a pivotal role during the supervised phase. It effectively mitigates the dissonance arising from sparse parameter selection. Additionally, as masks are computed prior to the supervised task and aggregated during the testing phase, the dual momentum mechanism proves **essential in both the supervised and unsupervised phases.**
> > >
> > > We provide results on the CIFAR100 dataset comparing two scenarios in **Table 1** below. Row 1 depicts the case where there is a single momentum in the supervised phase and dual momentum in the unsupervised phase. A lower performance highlights the importance of dual momentum for the supervised phase as well. The second row highlights the case where we directly copy the frozen parameters of the student to the teacher and let other parameters get updated by the usual EMA. This gives worse performance since it accelerated the updates of the frozen params, which should have been slower.
> > >
> > > | Method                                                                                     | Acc, F. |
> > > |--------------------------------------------------------------------------------------------|----------------------------------|
> > > | DoSAPP with single momentum during supervised phase and dual momentum during unsupervised phase | 71.93, 15.89                   |
> > > | DoSAPP with directly copying frozen (m=0) parameters from student to teacher               | 67.63, 22.46                   |
> > > | **DoSAPP (from paper)**                                                                    | **79.16, 7.73**                    |
> > >
> > > **Table 1**: Results comparing different scenarios on CIFAR100 dataset. We report the average accuracy(Acc.) and forgetting(F.).

---

### Official Review · Reviewer_4MGu · 2024-10-31

**Soundness:** 3
**Presentation:** 1
**Contribution:** 3
**Rating:** 8
**Confidence:** 4

**Summary:**

The paper proposes to use the test time data to alleviate the forgetting problem in the class incremental learning problem. The proposed method includes two training phases: a supervised learning phase using labeled training data, and an unsupervised learning phase using test time data. With the proposed DoSAPP, the forgetting problem can be greatly alleviated, and thus achieve a promising final performance.

**Strengths:**

1. The paper is well-motivated.
2. The idea of using test-time data to alleviate forgetting is interesting, and the performance is attractive.
3. It is surprising to have a positive backward transfer, thanks to test time data.

**Weaknesses:**

**Major:**
1. It can be beneficial to have an ablation study about the size of test-time data $D^u$, with respect to the final performance.

2. It is suggested to include more information about the experiments. For example, the error bars of major experimental results in Table 2. Also, it is suggested to include detailed training time information (comparisons with other methods, training time for supervised and unsupervised phases)

3. The presentation of the paper is a concern. There are significant formatting issues/typos/non-completed equations/informal telegrapheses in the writing. A careful overhaul or English proofreading is suggested.

**Minor:**
1. It can be interesting to expand the experiments from pretrained CLIP to conventional classification backbones, like ImageNet-21k pretrained encoders and FC classifiers.

2. The equations should also be properly punctuated.

**Summary:**

The paper presents an attractive setting and a novel view of alleviating forgetting in the CIL problem. However, there are significant issues, which limit the readability and the reproducibility, and thus I lean towards rejection. I'm happy to update my score if the concerns are properly addressed.

**Questions:**

Please refer to the weakness section.

---

> ### Author Response · Authors · 2024-11-23
> **Response to 4MGu**
>
> Dear reviewer 4MGu, thank you for the review and constructive questions/suggestions! Here is our response to your questions and concerns.
>
> # Addressing Major Weaknesses
>
> ## 1. Ablation study about the size of test-time data $D^u$
> We conducted an ablation with the size of $D^u$, for the CIFAR100 dataset and added it to the Appendix(A.4) in the revised rebuttal submission. In our method, we divided the evaluation data into two halves. One half is for unsupervised learning ($D^u$), and the other half is for evaluation ($D^e$). In the table below, we feed the fraction of $D^u$ for test time learning. $0.25$ means that $25\%$ of the original $D^u$ is fed to the model for unsupervised learning. We notice that when the fraction is below $0.75$, there is an appreciable difference between the performance of our proposed model. However, at $0.75$, the performance is quite close to that of the whole $D^u$.
> | Fraction of Test Data ($D^u$) | Acc, F         |
> |-----------------------------|----------------|
> | 0.25                        | 73.97, 14.23   |
> | 0.5                         | 76.83, 9.44    |
> | 0.75                        | 79.02, 8.16    |
> | 1                           | 79.16, 7.73    |
>
> ## 2. More information about the experiments
> 1. We have added the error bars in Table 2 for our method DoSAPP. We have not included the error bars for the baselines, for better readability.
>
> 2. We conducted a time analysis for our method as requested by the reviewer on a single batch of supervised training phase(on $D^t$) and unsupervised training phase(on $D^u$) for CIFAR100 dataset with a batch size of 64.
>
>    - Training Time per batch: 0.178 ms
>
>    - Unsupervised Training Time per batch: 0.173 ms
>
>    - Inference Time per batch: 0.012 ms
>
> ## 3. Addressing the formatting issues
> We thank the reviewer for pointing out the typos. We have fixed them all, and further revised for formatting issues in the revised rebuttal submission.  Please let us know if there are any additional concerns.
>
> # Addressing Minor Weaknesses
>  ## 1. Extending our setup to conventional classification backbones, like ImageNet-21k pretrained encoders and FC classifiers
>
> In this work, we focused on foundational models in particular CLIP since it has shown more generalisability than other pretrained models. This can be attributed to the absence of classification heads which are one of the causes of severe catastrophic forgetting in other pretrained architectures with FC classifiers. We follow the experimental standards as given in  SPU[a], FinetuningCLIP[b]. Moreover, given the short span of rebuttal, extending this work to pre-trained ImageNet-21k encoders and FC classifiers is currently not possible, and can be considered as future work.
>
>
> [a] Overcoming Generic Knowledge Loss with Selective Parameter Update by Zhang et al(CVPR 2024)
>
> [b] Finetune like you pretrain: Improved finetuning of zero-shot vision models by Goyal et al(CVPR 2023)
>
> ## 2. Fixing punctuation of equations
> We have fixed those issues in the revised rebuttal submission.

---

> > ### Comment · Reviewer_4MGu · 2024-11-25
> > **Thank you for your rebuttal**
> >
> > I thank the authors for their effort in the rebuttal. It has addressed my concerns and problems. Also, since the author promised to improve the readability, I would like to raise the score by 1.
> >
> > I have a further question. As referred from Fig. 1, some test time data are reused in the TTL stage. For example, the test data in **Task 1** is used in TTL at the end of the training on **every task**. This might weaken the CIL setting because in the TTL stage, it uses all test time data in **all** previous tasks. If this is the case, I'm considering maybe it's fairer to conduct TTL **only once** before each evaluation (i.e., for performance after training on task *i*: train the model with normal protocol from task *1* to task *i*, then conduct TTL only once on task *i* using all previous test data, without TTL on task *1* to task *i-1*).
> >
> > I understand that given the short time it's hard to address this in the rebuttal period, but I believe this can be important. And if I were correct, it might be interesting to have such a setting in the revised manuscript.

---

> > > ### Author Response · Authors · 2024-11-25
> > > **Response to 4MGu**
> > >
> > > We sincerely thank you for appreciating our efforts and for increasing the score. We are also glad that we were able to adequately address your concerns.
> > >
> > > Regarding your insightful observation about the TTL stage and its potential impact on weakening the CIL setting, we would like to emphasize that in a production setting, where models must adapt as new classes are introduced, test data from previously learned tasks is often abundant but underutilized, serving only for evaluation. We leverage this test data with our method, DoSAPP, to mitigate forgetting through unsupervised learning. To make this setting more practical, we perform Test Time Learning in an online fashion. We think that utilizing the test data from previous tasks repetitively does not weaken the CIL setting, because at test time, there is no information about tasks in the test data $D^u$. It's randomly shuffled.
> > >
> > > We still conduct an experiment on all the datasets where Test Time Learning happens only once after finishing all the tasks since it shall give a more fair comparison. We notice that there is a small drop in performance to our original proposed pipeline since now the model has no opportunity to rectify itself during test time in between the tasks. We also observe that despite the performance drop, DoSAPP still outperforms the majority of baselines in Table 2,
> > >
> > > | Method                              | Cifar100         |     Aircraft      |     Cars         | GTSRB        | Cub          |
> > > |---------------------------|----------------|-----------------|--------------------|------------------|--------------|
> > > | DoSAPP with TTL only once in the end  | 74.84 , 13.17|  36.79, 18.23   | 72.43, 1.05 | 70.13, 3.62 | 65.87, 4.34 |
> > > | **DoSAPP (from paper)**                | **79.16, 7.73** | **39.14, 12.55**  | **74.87, -0.74** | **72.33, 1.02**  | **68.17, 2.15** |
> > >
> > > Table-A Accuracy and Forgetting comparing DoSAPP with TTL only applied after completing the training of all tasks for different datasets.

---

> > > > ### Comment · Reviewer_4MGu · 2024-11-26
> > > > **Thank you for your information**
> > > >
> > > > I would like to thank the authors for the effort in the rebuttal period.
> > > >
> > > > I do agree that "test data from previously learned tasks is often abundant but underutilized" in practice, and the online adaptation makes a lot of sense. Also, thank you for sharing the extra experiments given the limited time, it helped in grounding the claim in the paper.
> > > >
> > > > I believe most of my concerns were properly addressed during the rebuttal, and the *only* concern now is about the presentation as we mentioned. I believe this work is valuable to the continual learning community, and thus I would like to raise my score to 8. However, I believe the paper needs **careful proofreading**, and also I strongly recommend the authors consider including the discussion in the rebuttal period when revising the paper's future version.

---

### Official Review · Reviewer_2tLK · 2024-11-04

**Soundness:** 4
**Presentation:** 4
**Contribution:** 3
**Rating:** 8
**Confidence:** 4

**Summary:**

This paper proposed to leverage the test-time data in CL to mitigate forgetting, which is novel, interesting and realistic. The authors further proposed a student-teacher learning framework to effectively learn and retain knowledge in this scenario. Specifically, the proposed DoSAPP(Double Smoothing via affine projected parameters) method consists of two stages: a supervised learning stage as in conventional CL and an unsupervised adaptation stage for test-time adaptation. In the former, the student would selectively update its parameters based on the gradient magnitude, and the teacher is a weighted exponential smoothing of the student model with dual momentum. In the latter, they first pseudo-label the test-time data based on the confidence of teacher and student models and then apply a similar learning paradigm as in the supervised learning stage. The authors validate the effectiveness of the proposed method in various datasets.

**Strengths:**

1. The proposed test-time adaptation to mitigate forgetting is novel, interesting and realistic.

2. The paper is easy-to-follow, well presented

3. The proposed method is simple but effective

**Weaknesses:**

Some experimental comparisons and ablation studies are missing and some additional details need to be added. Please refer to the Questions section.

**Questions:**

1. Are the comparisons with existing works fair?

Existing work such as L2P, DualPrompt, SLCA utilized a pretrained ViT model, i.e. ViT-B/16, which is trained on classification tasks, whereas the proposed method leveraged CLIP-ViT-B/16, which is trained on image and text pairs. Did the authors repeat experiments of existing methods with the same backbone? If the backbones are different, it is not convincing to conclude the superiority in performance in Table 2. In addition, the learning strategy of existing methods is also quite diverse: L2P and DualPrompt are prompt-based CL methods, whose backbone is frozen, and SLCA fine-tunes the backbone, ZSCL leverages external unlabeled datasets for finetuning. It seems unreasonable to put all these methods together and directly compare them with DoSAPP.


2. Why is the ablation study of single momentum + mask union missing in Table 3?

Although it was illustrated in lines 462-464 in the text, the scores are not presented in the table.

3. How is the self-labeling (line 422) done?

It seems that the authors used the fine-tuned model (line Finetune in Tab 2) to do pseudo-labeling, is that correct? Even though this seems a valid choice, an intuitive adaptation of existing methods to the test-time adaptation scenario is to use existing methods, e.g. SPU or DualPrompt, and use their trained model after each task to perform pseudo labeling on test data and further adapt these models as test-time adaptation (TTA). That might be a more comparable baseline in this novel scenario.

4. Some small typos need to be corrected

line 215 'were' -> 'where'

table 2, Acc. of CIFAR100, ZSCL should be bold instead of DoSAPP, similarly, Acc of GTSRB should be L2P, F. of GTSRB should be SLCA.

**Details Of Ethics Concerns:**

Looks ok.

---

> ### Author Response · Authors · 2024-11-22
> **Response to 2tLK**
>
> Dear Reviewer 2tLK, thank you so much for the review and the constructive suggestions! Here is our response to your questions and concerns.
>
> # Fairness of comparison
> 1. Yes the comparison is fair since we repeated the experiments for all the given baselines in Table-2 with CLIP ViT-B/16 backbone. We have added this clarification in the Table-2’s caption. (line 391 in the revised submission)
>
> 2. Since our proposed method is a novel contribution of improving class incremental continual learning(CIL), we compared our method with the most recent and diverse CIL algorithms.  We consider the diversity of approaches to the same problem a strength of the benchmarking.
>
> # Added the missing ablation of single momentum + union of mask.
> We thank the reviewer for pointing this out. We have now added the ablation(3rd row) in Table 5 of the revised rebuttal submission. The new results as compared to DoSAPP are also shown below. For each dataset, average accuracy and forgetting(separated by comma) is shown below,
> | Method                              | Cifar100    | Aircraft     | Cars        | GTSRB       | Cub         |
> |-------------------------------------|-------------|--------------|-------------|-------------|-------------|
> | Single Momentum + Mask Union        | 72.66, 8.86 | 31.79, 10.42 | 70.99, 3.64 | 61.54, 4.01 | 66.98, 3.17 |
> | **Dual Momentum + Mask Union (paper)**  | **79.16, 7.73** | **39.14, 2.55**  | **74.87, -0.74** | **72.33, 1.02** | **68.17, 2.15** |
>
> # Explanation of the use of Self Labelling
>  In Table 2, for self-labeling, a finetuned model is used as correctly observed by the reviewer. To have a more comprehensive comparison, we compare our method with naive integration of most recent CL method(SPU) with pseudo labeling in Table 5. We also want to emphasize that although most methods from the TTA literature are not directly applicable to our scenario, we have now added a SOTA TTA approach RMT[a] to SPU and SparsCL in the revised rebuttal submission(Table 5) and observed that the TTA method actually degrades the performance, as shown in the table below. This highlights that TTA methods which are often designed for input distribution shifts that don't affect the labels do not apply well for our setting where the unsupervised data is coming from previous task data distributions.
>
> | Method                              | Cifar100         |     Aircraft      |     Cars         | GTSRB        | Cub          |
> |---------------------------|----------------|-----------------|--------------------|------------------|--------------|
> | SPU + Pseudo Labeling| 74.09, 10.43| 27.72, 24.86  | 68.91, 7.34  | 60.17, 6.94  | 61.21, 4.01  |
> | RMT + SPU                           | 63.06, 23.28 | 29.33, 15.10  | 62.32, 21.95 | 54.13, 17.56 | 63.87, 6.34  |
> | RMT + SparseCL                      | 70.82, 12.25| 27.11, 16.29  | 69.81, 17.22 | 51.98, 11.40 | 60.03, 10.58 |
> | **DoSAPP (from paper)**                | **79.16, 7.73** | **39.14, 12.55**  | **74.87, -0.74** | **72.33, 1.02**  | **68.17, 2.15** |
>
> **Table-A** Accuracy and Forgetting for CIL methods integrated with RMT compared with DoSAPP over different datasets.
>
> [a] Robust Mean Teacher for Continual and Gradual Test-Time Adaptation by Döbler et al(CVPR 2023)
>
> # Corrected the typos:
> We have corrected all the typos, bold font usage in tables, and proofread all the text once more in the revised rebutall submission. Thank you so much for pointing them out.

---

> > ### Comment · Reviewer_2tLK · 2024-11-24
> > **Further  questions**
> >
> > Thank you for your clarification. I especially appreciate the experiments with state-of-the-art TTA methods. I consider my concerns about ablation and self-labeling to be fully addressed.
> >
> > However, I have an additional question about the comparison in Table 2. As the authors mentioned in the previous response, all these methods use the CLIP ViT-B/16 backbone. When I compare the reported performance in this submission with that reported in previous works, e.g. SLCA, I observe a significant performance drop in the same dataset, e.g. CUB200 or CIFAR-100. For instance, SLCA (with IN1k-Self backbone) has 73.01% in their paper whereas it has only 53.87% of average accuracy in your submission, for 10-task CUB200 experiments. Could the authors elaborate more on this? Is this purely due to the backbone change or might there be some sub-optimal hyperparameters?

---

> > > ### Author Response · Authors · 2024-11-25
> > > **Response to 2tLK**
> > >
> > > Thank you for appreciating our efforts with the state-of-the-art TTA experiments. We are grateful that our response addressed your concerns about ablation and self-labeling.
> > >
> > > Regarding your observation about the performance discrepancy in Table 2, we would like to clarify that methods such as SLCA are indeed highly sensitive to the choice of the pre-trained backbone. For example, as highlighted in the SLCA paper (Table 2), there is a notable accuracy gap of approximately 10% between supervised pretraining on ImageNet-21K and self-supervised pretraining on ImageNet-1K using MoCo v3. This trend is further illustrated in Figure 3 of their work. Additionally, it is important to consider that some supervised tasks share significant similarities with ImageNet-21K's supervised task, which naturally leads to exceptionally high performance on datasets like CIFAR-100.
> > >
> > > Finally, as seen from the SPU results, we observe that methods like SLCA encounter considerable challenges when scaling with CLIP backbones, which underscores their sensitivity to backbone selection. We sincerely hope this explanation addresses your query and are happy to provide further clarifications if needed.

---

> > > > ### Comment · Reviewer_2tLK · 2024-11-25
> > > >
> > > > Thanks for your clarification. I confirm that all my questions are fully addressed by the authors and I believe this is an interesting work that deserves acceptance in this venue. Therefore, I will raise my rating to 8.

---

> > > > > ### Author Response · Authors · 2024-11-25
> > > > > **Response to 2tLK**
> > > > >
> > > > > We sincerely thank you for your thoughtful review and for recognizing the contributions of our work. We deeply appreciate your engagement with our paper, your constructive feedback, and your positive assessment. Thank you so much for your raising your score to 8.

---

### Meta-Review · Area_Chair_igw7 · 2024-12-25

**Metareview:**

The paper received quite diverse reviews. While reviewers 2tLK and 4MGu have given high scores, other two reviewers don't seem to prefer clear acceptance of this work. Upon reading myself the paper carefully and considering the points during discussions with other reviewers, unfortunately I'm recommending rejection of this work in its current form. Primary reasons for this rejection (most have already been discussed during the rebuttal) are mentioned below.

The proposed intuition and the method presented in the paper is **highly promising** and would strongly suggest the authors to incorporate the comments below to make it a solid submission in the future.

Reasons for rejection:

---
- While the broad idea of utilizing test samples to mitigate forgetting seems promising, it is not entirely clear what led to the said design choices for TTA and why other CL methods do not benefit from it (reviewers uBk1 and PpH5 shared similar concerns). Though many questions relating this have been answered during the rebuttal, I (and other reviewers) however believe that the paper requires **serious rewriting** to ensure smooth and logical connection between thoughts.

- Limited experiments. Current set of experiments are only performed using CLIP ViT-B/16 (similar to SPU). Since other CL baseline methods do not benefit from TTA (as mentioned by the authors during rebuttal), it becomes even more important that the proposed method is shown to perform well on a variety of settings to ensure its generality. For example:
	- CLIP with RN50, ViT-B/32
	- FLAVA with ViT-B/16
	- vision only foundation models such as DINO and Masked Auto-Encoders

- Comparing the method with other baselines (involving large pre-trained models) would be valuable, for example RanPAC or even recently proposed continual fine-tuning methods.

- It would also be beneficial to discuss situations where both the student and the teacher are wrong during inference. Highly likely in open-world setting where test samples do not necessarily belong to the train distribution. In this case, other methods wouldn't suffer as they don't use test samples for adaptation, however, the proposed method might suffer. A discussion would be useful.

[1] RanPAC: Random Projections and Pre-trained Models for Continual Learning

**Additional Comments On Reviewer Discussion:**

- Main concerns raised by the reviewers were regarding (1) self-labeling; (2) lack of ablation; (3) writing quality; (4) lack of comparisons (experiments); (5) justification behind dual momentum etc.
- The reviewers greatly appreciated the active engagement of the authors during the rebuttal period. Several new results were provided by the authors (e.g., ablation of single momentum, test data size etc.) in order to show the effectiveness of their approach.
- Most of the replies from the authors were convincing however, the fact that there were several doubts and questions regarding the work did leave an impression that the manuscript needed a very careful rewriting to logically connect thoughts with justifications to make it readable.
- There were other concerns as well regarding the validity of the approach (mentioned above) which we believe requires careful attention.

---

### Decision · Program_Chairs · 2025-01-22

Reject